# Preventive and treatment efficiency of dendrosomal nano-curcumin against ISO-induced cardiac fibrosis in mouse model

Behnaz Beikzadeh[1], Mona Khani[1], Yasamin Zarinehzadeh[1], Elham Abedini Bakhshmand[1], Majid Sadeghizadeh[1], Shahram Rabbani[2], Bahram M. Soltani[1]*

1 Genetics Department, Faculty of Biological Sciences, Tarbiat Modares University, Tehran, Iran,
2 Research Center for Advanced Technologies in Cardiovascular Medicine, Cardiovascular Diseases Research Institute, Tehran University of Medical Sciences, Tehran, Iran

* soltanib@modares.ac.ir

**Data Availability Statement:** All relevant data are within the manuscript and its Supporting Information files.

## Abstract

Cardiac fibrosis (c-fibrosis) is a critical factor in cardiovascular diseases, leading to impaired cardiac function and heart failure. This study aims to optimize the isoproterenol (ISO)-induced c-fibrosis model and evaluate the therapeutic efficacy of dendrosomal nano-curcumin (DNC) in both in-vitro and in-vivo conditions. Also, we were looking for the differentially expressed genes following the c-fibrosis induction. At the in-vitro condition, primary cardiac fibroblasts were exclusively cultured on collagen-coated or polystyrene plates and, were treated with ISO for fibrosis induction and post-treated or co-treated with DNC. RT-qPCR and flow cytometry analysis indicated that DNC treatment attenuated the fibrotic effect of ISO treatment in these cells. At the in-vivo condition, our findings demonstrated that ISO treatment effectively induces cardiac (and pulmonary) fibrosis, characterized by pro-fibrotic and pro-inflammatory gene expression and IHC (α-SMA, COL1A1, and TGFβ). Interestingly, fibrosis symptoms were reduced following the pretreatment, co-treatment, or post-treatment of DNC with ISO. Additionally, the intensive RNAseq analysis suggested the COMP gene is differentially expressed following the c-fibrosis and our RT-qPCR analysis suggested it as a novel potential marker. Overall, our results promise the application of DNC as a potential preventive or therapy agent before and after heart challenges that lead to c-fibrosis.

## Introduction

A significant pathophysiological issue in cardiovascular disease (CVD), is the fibrosis incidence, which could lead to impaired cardiac function and an increased risk of heart failure. Cardiac fibrosis types vary *in* cause and location and are categorized into four groups: replacement fibrosis, reactive interstitial fibrosis, infiltrative interstitial fibrosis, and endomyocardial fibrosis [1–4]. According to previous studies, diabetes, COVID-19, obesity, and ageing can also contribute to cardiac fibrosis [5, 6]. Fibrosis occurs in various chronic diseases and shares common molecular and cellular processes in multiple organs, such as the heart, liver, lungs,

**Funding:** This work is based upon research funded by Tarbiat Modares University and Iran National Science Foundation (INSF) under project No.4014277. The funders had no role in study design, data collection and analysis, decision to publish, or preparation of the manuscript.

**Competing interests:** The authors have declared that no competing interests exist.

and kidneys, resulting in organ impairment [7–9]. Pulmonary fibrosis is an inflammatory and chronic lung disease with many causes, such as rheumatoid arthritis, lupus, COVID-19, environmental factors such as smoking, and the effects of some drugs such as bleomycin. During this disorder, the alveoli walls thicken, disrupting gas transfer and causing respiratory failure [10].

Following exposure to various conditions, including oxidative stress, hypoxia, inflammation, death of cardiomyocytes and mechanical stress, the damaged tissue initiates a fibrotic response and produces extracellular matrix (ECM) proteins as a replacement for the lost cells to sustain its structural integrity [2, 5, 11]. In fibrosis, ECM proteins such as collagens (Particularly collagen I and III), elastin and fibronectin get excessively deposited at the site of the wound, which leads to stiffness, overgrowth, reduced contractions and signal transduction capacity, and consequently, abnormal cardiac function [4, 12, 13]. The principal source for the deposition of ECM components in the damaged heart is myofibroblasts, which differentiate from Cardiac fibroblasts (CFs) [14]. Despite comprising about 10% of the total heart volume, CFs are one of the most considerable cell populations in the heart [15]. CFs produce extracellular matrix proteins, provide structural support in the cardiac tissue, and mediate tissue homeostasis by participating in autocrine and/or paracrine pathways [16, 17].

There are several signaling pathways involved in the development of fibrosis including, Renin–Angiotensin–Aldosterone System (RAAS), Endothelin-1 (ET-1), transforming growth factor-beta (TGF-β), Wnt/β-catenin, and GPCR/Adrenergic pathways. Inflammatory and profibrotic factors that regulate fibrotic response are as follows tumor necrosis factor-α (TNF-α), interleukin [18]-1, IL-6, IL-10, and IL-4, connective tissue growth factor (CTGF), platelet-derived growth factors (PDGF) and matrix metalloproteinases (MMPs) which come from multiple cells such as necrotic cardiomyocyte and inflammatory cells, and could directly or indirectly increase the production of ECM by myofibroblasts [5, 12, 18–20]. In the context of differentiation, fibroblasts undergo changes in cellular morphology (e.g. developing a contractile form, mitochondrial alterations, extension in the endoplasmic reticulum) and also in gene expression patterns such as upregulation of ECM proteins (e.g. collagen I, collagen III and collagen VI), fibronectin, and EDA splice variant of fibronectin, alpha-smooth muscle actin (α-SMA), TGF-β and many other profibrotic factors [11, 14]. The release of signaling molecules by myofibroblasts and the trans differentiation of other cell types to myofibroblasts further contribute to the inflammatory response and augment myofibroblast populations through positive feedback loops [21].

Given the significant role of myofibroblasts in fibrosis and limited therapies, we attempted to Comprehensively investigate fibrosis by generating both in vitro and in vivo fibrotic mouse models using isoproterenol (ISO) treatment, and examined the effect of a fibrosis deactivator, dendrosomal nano-curcumin (DNC). Differentiation of primary fibroblasts commences within days of isolation, even in the absence of chemical stimulators; therefore, CFS should be cultured in substrates of physiological stiffness. Culturing in collagen-coated plates provides the required elastic modulus of a healthy myocardium (*E* of ~7 kPa) and helps to maintain CFs without considerable differentiation into myofibroblasts [22, 23]. We introduced a standardized protocol for inducing cardiac fibrosis in mice using ISO treatment. In addition to inducing cardiac fibrosis, the possibility of inducing pulmonary fibrosis using ISO was investigated. Furthermore, we studied the effects of the DNC on the generated models. Studies have shown that curcumin has anti-fibrotic effects in various disorders in the heart, liver, and lungs [24–26]. Dendrosomal nanoparticles have been used to improve the therapeutic potential of Curcumin [27].

Our results in utilizing combinations of treatments for progression and alleviation of fibrosis will help to discover targeted therapeutic approaches to assist patients with cardiac fibrosis. Additionally, we present a hub gene (COMP) using RNA-seq data analysis, that is upregulated during cardiac fibrosis and may contribute to cardiac dysfunction.

## Material and methods

### Ethics statement

This study was approved by the Research Ethics Committees of Tarbiat Modares University (IR.MODARES.REC.1400.363). According to the Guide for the Care and Use of Laboratory Animals published by the U.S. National Institutes of Health (NIH Publication No. 85–23, revised 1996), animal experiments were conducted.

### Primary cardiac fibroblast culture

Primary cardiac fibroblasts were extracted from 3-day-old neonatal C57BL/6 mice. Briefly, mouse hearts were washed 3 times with cold phosphate-buffered saline (PBS) containing 1% Penicillin and Streptomycin (Pen/Strep, BioIdea Co, Tehran, Iran) and then the hearts were minced into 1 $mm^3$ sections. Cardiac fibroblasts were isolated by enzymatic digestion with Trypsin-EDTA (1X) 0.25% (BioIdea Co., Tehran, Iran) and collagenase type IV (Cat #17104019, Gibco) (2:1) in a shaker incubator at 37˚C for 15 minutes. Digested cells were collected from the supernatant in Fetal bovine serum (FBS, Gibco). The digestion process was continued until the heart tissues were completely digested (5–6 cycles). Eventually, the supernatants present in FBS were centrifuged at 300g for 5 minutes. The cell pellet was resuspended in DMEM/F12 (BioIdea Co, Tehran, Iran) with 10% FBS, 1% Pen/Strep, and 1% L-glutamine (Gibco) and then distributed in the T25 flask and Softwell 12 with 4 kPa collagen gel stiffness (Matrigen Softwell) for 3 hours in the incubator at 37˚c with 5% CO2. After 3 hours, the supernatant containing unattached cells was removed and attached cardiac fibroblasts were washed 3 times with PBS; Meanwhile, fresh complete media was added. After adopting a spindle-shaped morphology, the cultured medium with 5% FBS was utilized [23, 28–30]. The first and second passages of cardiac fibroblasts were used in the current study. PCR tests for markers such as vimentin and fibronectin were conducted to confirm the fibroblasts, as well as the examination of α-SMA expression as the gold standard for identifying activated fibroblasts. Additionally, MYH11 PCR testing was performed, and no MYH11 gene-related band was observed, indicating the absence or non-detectable number of VSMCs in our cultured cell preparation.

### MTT assay

To determine an optimal DNC treatment dose, cell viability was assessed using the MTT test. Primary cardiac fibroblasts were cultured overnight at a density of 5×103 cells in 96-well plates. Nano-curcumin was purchased from Alborz Nano-drug Tech in Iran and optimized protocol was described previously [31]. DNC was used with different doses (1μm, 5μm, 10μm, and 15μm) for cardiac fibroblast treatment. After passing 24 hours, the cells were washed with PBS and then incubated with MTT solution (5mg/ml) (Sigma-Aldrich, USA) for 4 hours and 200μl of dimethyl sulfoxide (Sigma-Aldrich, USA) for 15 minutes in 37˚c, respectively. The OD was measured at 570 nm with a spectrophotometer (BioTek, USA). The cell viability percentage was calculated using this formula: Cell Viability = (Mean $OD_{sample}$ - blank absorbance/Mean $OD_{control}$ - blank absorbance) ×100.

### ISO and DNC treatment

Cardiac fibroblast cells were seeded at a density of 20×$10^4$ per well of a 12-well plate overnight. The cells were divided into untreated, ISO (i5627/Sigma-Aldrich), ISO/DNC (ISO for 24 hours and then DNC treatment), ISO+DNC (co-treatment) and, DNC groups. The cardiac fibroblast cells were treated with ISO (10μm) [30, 32] and DNC (1μm) for 24 hours. At the

same time, the cells were cultured in the 4 kPa Softwell plate to see the difference between the cardiac fibroblast cells inside the collagen plate and the usual polystyrene plates.

## Cell cycle analysis with flow cytometry

After treating cardiac fibroblasts with ISO and DNC, cells were trypsinized with trypsin-EDTA (0.25%) for 5 minutes. Subsequently, cells were pelleted at 300g for 5 minutes through the centrifuge and washed with PBS. The fixation stage was done on cell pellets using 70% cold ethanol for 2 hours afterwards cellular DNA was stained in PBS containing Propidium iodide (Sigma-Aldrich, USA) for 30 minutes out of light. Cell cycle analysis was conducted on a BD FACS Calibur Flow Cytometer (BD Bioscience, USA).

## Animal model designs for cardiac fibrosis

This experimental study used C57/BL6 male mice (22±1g) (3 per group). The mice were obtained from the Laboratory Animal Breeding Center of the Veterinary Medicine Faculty at Tehran University. Before the experiments, the animals were kept under a pathogen-free condition and fed regular food and water. They were maintained under standard lighting conditions (12-h light/dark cycle), temperature (20–22°C), and humidity (50–60%) for 7 days.

As there is no standard dose for inducing cardiac fibrosis via ISO, different doses of this drug were used to determine the most suitable dose for the experiment. In the first group, ISO was injected subcutaneously for 14 days (10 mg/kg for 3 days and 5 mg/kg for 11 days), and on the 15[th] [33, 34], and the 25th days, the mice were sacrificed. In the second group, ISO was injected subcutaneously with a dose of 5 mg/kg/day for 11 days, and then on the 12th day, the mice were sacrificed [35–37]. In the third group, ISO was used subcutaneously with a dose of 10 mg/kg/day for 11 days, and then on the 12th day, the mice were sacrificed. Finally, considering the usage of a dose of 170 mg/kg for inducing heart failure with isoproterenol [38], a lower dose (140 mg/kg) was chosen and studied due to its minimal mortality in mice compared to the higher dose (Table 1). Twenty-one mice were randomly assigned to seven groups (three per group) for treatment, namely, control and ISO. Isoproterenol (140 mg/kg) was subcutaneously injected for four consecutive days to induce cardiac fibrosis, and the mice were sacrificed on days 5, 7, 11, 18, 25, and 32 through cervical dislocation (Fig 1). The mice in the control group also received the same volume of saline. The hearts were harvested for additional molecular biological and pathological investigations.

## Histopathological examination

**Tissue sampling and preparation.** Heart tissues were harvested from ISO-treated and control mice on days 5 and 32. The tissues were promptly fixed in 10% neutral buffered formalin (NBF) for 24 hours. Subsequently, the tissues underwent meticulous procedures: initial dehydration through an ethanol gradient, followed by paraffin embedding, and ultimately

**Table 1. Protocols used for cardiac fibrosis induction.**

| Protocol | ISO- injection protocol | Harvesting days |
|:---:|:---:|:---:|
| 1 | 3 days (10 mg/kg) + 11 days (5 mg/kg) inject | 15[th] or 25[th] |
| 2 | 11 days (5 mg/kg/day) inject | 12[th] |
| 3 | 11 days (10 mg/kg/day) inject | 12[th] |
| 4 | 4 days (140 mg/kg) inject | 5, 7, 11, 18, 25, 32[th] |

Protocols #1, 2, and 3 are adapted from the literature and protocol #4 is novel and recommended.

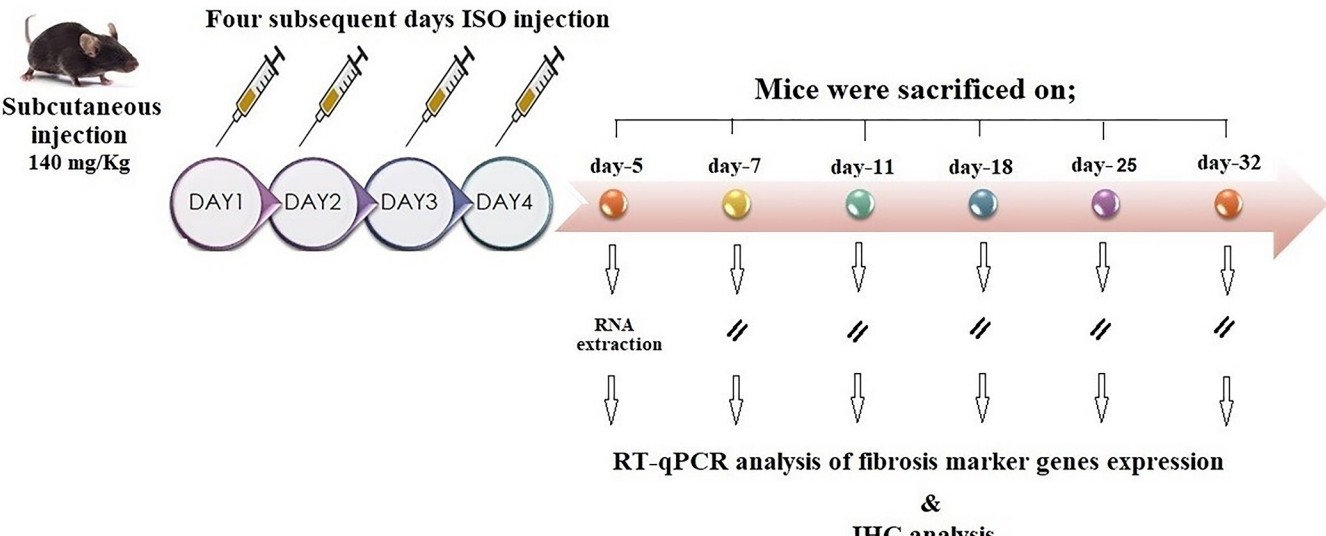

**Fig 1. Outline of the study protocol.** The mice model was injected subcutaneously for four sequential days, and then 3 mice were harvested at days 5, 7, 11, 18, 25, and 32 either for the RNA extraction or IHC analysis.

sectioning into 5-μm thick slices. These slides served as the foundation for subsequent two-step immunohistochemistry (IHC), hematoxylin and eosin (H&E) staining, and Masson trichrome staining (MTS). The acquired images were then subjected to quantification and analysis utilizing ImageJ software (Version 1.53t, 24 August 2022).

**Hematoxylin and eosin staining.** H&E staining analysis was performed to observe the alterations in the morphology and organization of cardiomyocytes and the structural characteristics of their nuclei. Staining was carried out utilizing a hematoxylin and eosin staining kit (ab245880).

**Masson trichrome staining.** In order to detect the presence of fibrosis within the tissues, MTS was conducted, adhering to the prescribed instructions provided by Sigma-Aldrich kit. This staining method enabled the visualization of collagen fibers in blue, cell nuclei in dark brown to black, and healthy regions in red. The acquisition of sample images was accomplished using a brightfield microscope, and subsequent quantification of the images was performed utilizing ImageJ software (Version 1.53t, 24 August 2022), enabling the analysis of the proportion of collagen area within the myocardial interstitial relative to the total field area.

**Immunohistochemistry (IHC).** To localize COL1A1, α-SMA and TGF-ß, a two-step IHC analysis was performed. Before the staining stage, the slides underwent deparaffinization and rehydration following the established protocol. The slides were incubated for 2 hours at room temperature (RT) with 1% bovine serum albumin (BSA) in TBS. For antigen retrieval (AR), the slides were incubated in a 10 mM Sodium citrate buffer (pH = 6.0) containing 0.05% Tween 20 for 20 minutes at a temperature of 95˚C. Endogenous peroxidase activity was efficiently quelled using 0.3% $H_2O_2$, thereby effectively reducing background staining. The slides were subsequently exposed to a mouse anti-COL1A1 antibody (ab88147; Abcam, Lucerne, Switzerland), anti-α-SMA antibody (ab124964), and anti-TGF-ß antibody (ab215715) at a dilution of 1:200 for 1 hour at RT. Finally, the slides were treated with HRP-polymer secondary antibodies and the DAB Substrate Kit (ab64238).

## DNC treatment of cardiac fibrosis mouse model

For DNC treatment, the mice were divided into 4 groups (3 per group). In the first and second groups, the mice were pre-treated with DNC for 7 days, followed by co-treatment of ISO and DNC for 4 days (DNC/ISO+DNC). They were sacrificed on D5 and D32. In the third group, the mice were treated with ISO for 4 days, followed by DNC treatment for 7 days and sacrificed on D32. In the fourth group, the mouse was treated with ISO for 4 days and then from D25-D31 injected with DNC and sacrificed on D32. For all groups, 10 mg/kg DNC was administered intraperitoneally [39, 40].

## Animal model designs for pulmonary fibrosis

Besides inducing cardiac fibrosis, we also used ISO to investigate the possibility of inducing pulmonary fibrosis in mice. To induce pulmonary fibrosis, the ISO was initially administered subcutaneously and intraperitoneally separately using a protocol of 10 mg/kg for 3 days and 5 mg/kg for 11 days. The mice were sacrificed on the 15th day.

Finally, eighteen mice were randomly assigned to six groups (3 per group) for treatment, namely, control and ISO. Isoproterenol (140 mg/kg) was intraperitoneally injected for four consecutive days to induce pulmonary fibrosis, and the mice were sacrificed on days 5, 11, 18, 25, and 32 through cervical dislocation. The mice in the control group also received the same volume of saline. The hearts were harvested for additional molecular biological investigations.

## RNA extraction and RT-qPCR

Total RNA was isolated from cardiac fibroblasts and mouse heart tissues using Trizol reagent (Pishgam Biotech Co., Tehran, Iran). The overall quality and quantity of the extracted total RNAs were evaluated using agarose gel electrophoresis and NanoDrop (Thermo Scientific, USA). All total RNA samples were reverse transcribed into cDNA using reverse transcriptase (Thermo Fisher Scientific, USA) following the manufacturer's protocol. qPCR was performed to detect gene expression levels using SYBR Green qPCR Master Mix (Amplicon, Denmark) and the Step One™ Real-Time PCR System (Applied Biosystems, USA) in a 40-cycle reaction. The mRNA expression levels were normalized to $\beta_2M$ as the internal control. The relative expression level and fold change were calculated using the $2^{-\Delta CT}$, and $2^{-\Delta\Delta CT}$ methods respectively. The sequences of specific primers are provided in the S1 Table.

## Statistical analysis

Statistical analyses were conducted using GraphPad Prism 8. The data were presented as the mean ± SEM based on three mice in each group. An unpaired two-tailed Student's t-test was employed to compare the two groups. The one-way ANOVA was used to compare the means of more than two groups. A p-value of less than 0.05 was considered to define a statistically significant difference.

## Eligibility criteria and dataset selection

In the bioinformatics section, our goal was to identify a gene that plays a role in every cardiac fibrosis induction model, including cell culture conditions. In fact, our objective was not to find a gene specific to one particular cardiac fibrosis induction model. Instead, we aimed to identify a gene that is universally involved across all conditions. Given that we utilized samples from various studies that shared only cardiac fibrosis as a common factor, with different induction methods and species.

## RNA-seq datasets

The RNA-seq data has been obtained from NCBI's Gene Expression Omnibus (GEO) and reported with GEO Series accession numbers (GSE97358, GSE123018, GSE152250, GSE151834). GSE97358 includes 168 primary human cardiac fibroblast samples (84 control, 84 TGFB1-induced cardiac fibrosis at 24h). GSE123018 contains 8 Primary human cardiac fibroblast samples (4 control, 4 TGFB1-induced cardiac fibrosis at 24h). GSE152250 consists of 12 human cardiac fibroblast samples (6 control, 6 TGFB1-induced cardiac fibrosis at 24h and 48h). GSE151834 includes 40 adult C57BL/6 mice subjected to ischemic cardiac injury due to permanent ligation of the left anterior descending coronary artery, responsible for supplying blood flow to the left ventricular myocardium. The fibrotic scar tissue in the injured region (20 samples) and uninjured region (20 samples) were dissected from the same heart for RNA-seq. Illumina HiSeq platform was used for all expression profiling.

## RNA-seq processing

Galaxy a Web-based platform (https://usegalaxy.org/) was utilized for finding differentially expressed genes (DEGs). Primarily, through Faster Download and Extract Reads in the FASTQ format tool, raw reads were extracted from the SRA dataset (https://www.ncbi.nlm.nih.gov/sra/). Secondly, the quality of each read sequence was controlled using the FastQC tool. In the next stage, Trimmomatic and Trim Galore tools were used to eliminate adapter contamination and low-quality reads. Afterwards, to align the reads with the reference genome and to count the aligned reads, the HISAT2 and featureCounts tools were utilized, respectively. Finally, Differential expression analysis was determined using DESeq2. Filtration criteria for obtaining DEGs include P-adj$<0.05$, $\log_2(FC)>1$ for up-regulated genes and $\log_2(FC)<-1$ for down-regulated genes.

## Finding hub genes

To find the hub genes, firstly the Venn diagram was drawn (https://bioinformatics.psb.ugent.be/webtools/Venn/) to identify shared up-regulated and down-regulated DEGs between the mentioned 4 GEO Datasets. These common genes were visualized through a heat map generated using GraphPad Prism. Additionally, the common genes were subjected to functional enrichment analysis using the ToppGene portal (https://toppgene.cchmc.org/) with a P-value $< 0.05$ cutoff.

In the next stage, the protein-protein interaction (PPI) network was obtained using the STRING database (https://string-db.org/). This network was provided for both mice and humans by shared genes. The string outcome (source and target genes with combined score) was subjected to Cytoscape. Source genes are typically the primary genes under study. Target genes are those that interact with the source genes. Using the CytoHubba plugin in the Cytoscape, hub genes were obtained. The genes with the highest scores were identified as hub genes.

# Results

## Optimal dose for DNC treatment

The optimal DNC treatment dose for cardiac fibroblast was determined using the MTT assay (Fig 2A). Cardiac fibroblast cell viability decreased as DNC concentration increased after 24 hours compared to the control group. Since there is no significant difference in cell viability between the 1μm DNC and the control group, it was chosen as the optimal dose without any cell cytotoxicity. The other doses showed high cytotoxicity.

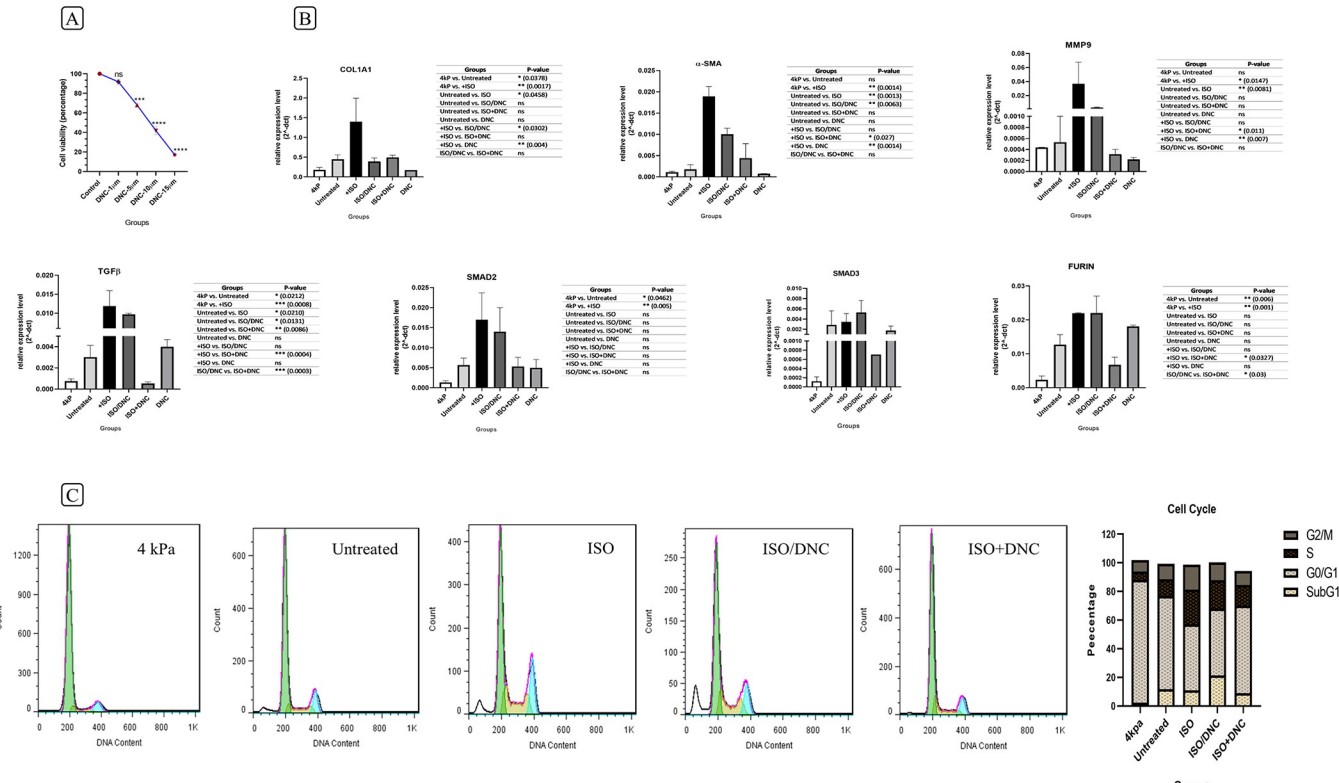

**Fig 2. Effect of DNC on cardiac fibroblasts activated by ISO.** (A) The optimal DNC treatment dose (1μm) for cardiac fibroblast without significant cytotoxicity was determined using the MTT assay. (B) Effects of ISO (10μm) and DNC (1μm) on profibrotic and pro-inflammatory gene expression were determined using RT-qPCR. Additionally, the differences between cardiac fibroblasts cultured on polystyrene plates (untreated) versus collagen plates (4 kPa) were reported. ISO and DNC are applied through sequential treatment (ISO/DNC) or co-treatment (ISO+DNC). Graph indicate that co-treatment of (ISO +DNC) is more effective for attenuation of fibrosis induction. (C) Cell cycle distribution of cardiac fibroblasts cultured in a softwell plate, and polystyrene plate namely untreated, ISO, ISO/DNC and ISO+DNC was examined using flow cytometry. Again, co-treatment of ISO+DNC is more effectively affecting cell cycle progression, shown through reduced S phase and increased G1 cell populations. Data are presented as mean ± SEM (triplicate). The mean expression is shown as relative expression level for RT-qPCR. T Student's t-test for MTT assay, One-way ANOVA test for RT-qPCR: ns: non-significance, *p<0.05, **p<0.01, ***p<0.001, ****p<0.0001.

## Effects of ISO and DNC on profibrotic and pro-inflammatory gene expression

High Pressure (>1000 kPa) in the common polystyrene plate causes fibroblasts to transform into myofibroblasts, indicating the significant impact of mechanical stress on cellular behavior (22). In the current study, a softwell plate with the 4 kPa stiffness was used despite the commonly used plates. As reported in Fig 2B, about some fibrosis markers (COL1A1, SMAD2, TGFβ and FURIN), there was a significant difference between cardiac fibroblasts in polystyrene plates in comparison to the collagen plate.

Using 10μM ISO in polystyrene plates increased fibrotic marker expression compared to the untreated group and cardiac fibroblasts cultured in softwell plates. Following the use of DNC, the expression of these markers decreased. A comparison of ISO + DNC co-treatment with separate ISO/DNC treatment shows a better reduction in fibrosis-related gene expression (Fig 2B). The two main fibrosis markers, COL1A1 and α-SMA, show significant changes among different groups.

### Effect of DNC on ISO-induced increase in the cell cycle S phase

Flow cytometry is a technique that allows us to precisely determine the proportion of cells in distinct stages of the cell cycle. This information is crucial to understanding the biological processes that are taking place in our samples. According to Fig 2C, it can be observed that the cardiac fibroblasts in common plates have a higher percentage of cells in the S (11.42%) and G2/M (10.98%) phases, a lower percentage of cells in the G0/G1 (65.18%) phase compared to the 4 kPa group (G0/G1 = 85.73%, S = 5.86%, G2/M = 7.85%). This indicates the mechanical effect of the polystyrene plates on the activation of the fibroblasts into myofibroblasts and their proliferation. On the other hand, induction of fibrosis using ISO has resulted in an increase in the number of cells in the S (24.24%) and G2/M (17.38%) phases compared to the untreated group as well as the cells cultured in the softwell plate. Following the treatment with DNC, the cell cycle of the ISO+DNC group was found to be closer to its normal state. There was a decrease in the S and G2/M phases by 14.49%, and 9.85% respectively, and an increase in the G phase by 60.05% as compared to the ISO group. This indicates a reduction in cell proliferation. The ISO/DNC group exhibited a decrease only in the S (19.87) and G2/M (12.31) phases compared to the ISO group.

### Literature-introduced protocols for the cardiac fibrosis induction

After doing protocol #1 (14 days injection of 10 & 5 mg/kg) in mice, RT-qPCR was used to evaluate the mRNA level of COL1A1 and α-SMA genes as the well-known biomarkers of cardiac fibrosis induction (S1 Fig). Although COL1A1 and α-SMA genes expression level was increased under these circumstances, that elevation was not significant on day 15 (S1A Fig). Interestingly, a reduction in the expression of these genes was observed 10 days after the last injection (day 25, which is not significant compared to the other related control groups). Alternatively, following the protocols of #2 & #3, namely, 11 days of subcutaneous injection of ISO at doses of 5 mg/kg or 10 mg/kg, did not significantly increase the expression of COL1A1 and α-SMA genes at day 12 (S1B Fig) Accordingly, protocols #1, #2, and #3 did not significantly induce cardiac fibrosis, indicating that they were not reliable.

### Reliable cardiac fibrosis induction shown by significant up-regulation of biomarker genes

In a novel protocol (#4), 140 mg/kg of ISO was administered in mice for 4 subsequent days. Then, the expression levels of pro-fibrotic and pro-inflammatory genes were determined by RT-qPCR on days 5, 7, 11, 18, 25, and 32. As reported in Fig 3, pro-fibrotic markers such as COL1A1, α-SMA, TGF-ß, SMAD3, FURIN, and pro-inflammatory genes like MMP9 showed a significant increase in their expression on day 5, compared to the control group. In addition, a group of markers such as α-SMA, TGF-ß, and MMP9 showed a significantly increased expression level on day 32 as well.

### Morphological changes in cardiomyocytes and collagen accumulation in the mouse model of cardiac fibrosis following the novel protocol

For morphological evaluation of cardiomyocytes and investigation of cardiac fibrosis levels, H&E staining of left ventricular tissue sections and Masson's trichrome staining (MTS) were performed, respectively. Using H&E staining (Fig 4A), degeneration and necrosis of cardiomyocytes were observed on D32, whereas no such morphological changes were observed in the control group and on D5. In the MTS staining (Fig 4B), the presence of cardiac fibrosis is documented through collagen accumulation. Accordingly, muscle fibers appear red, collagen

Fibrotic marker genes expression following 4 days of ISO injection

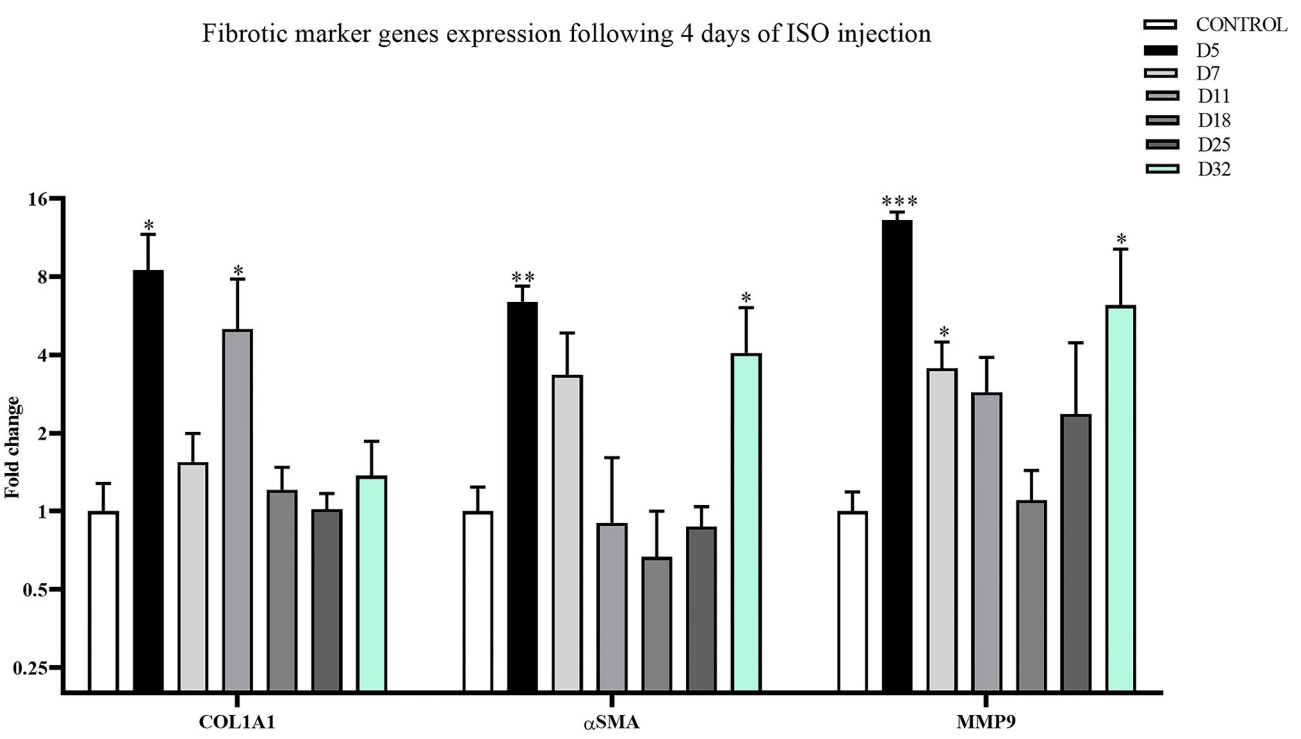

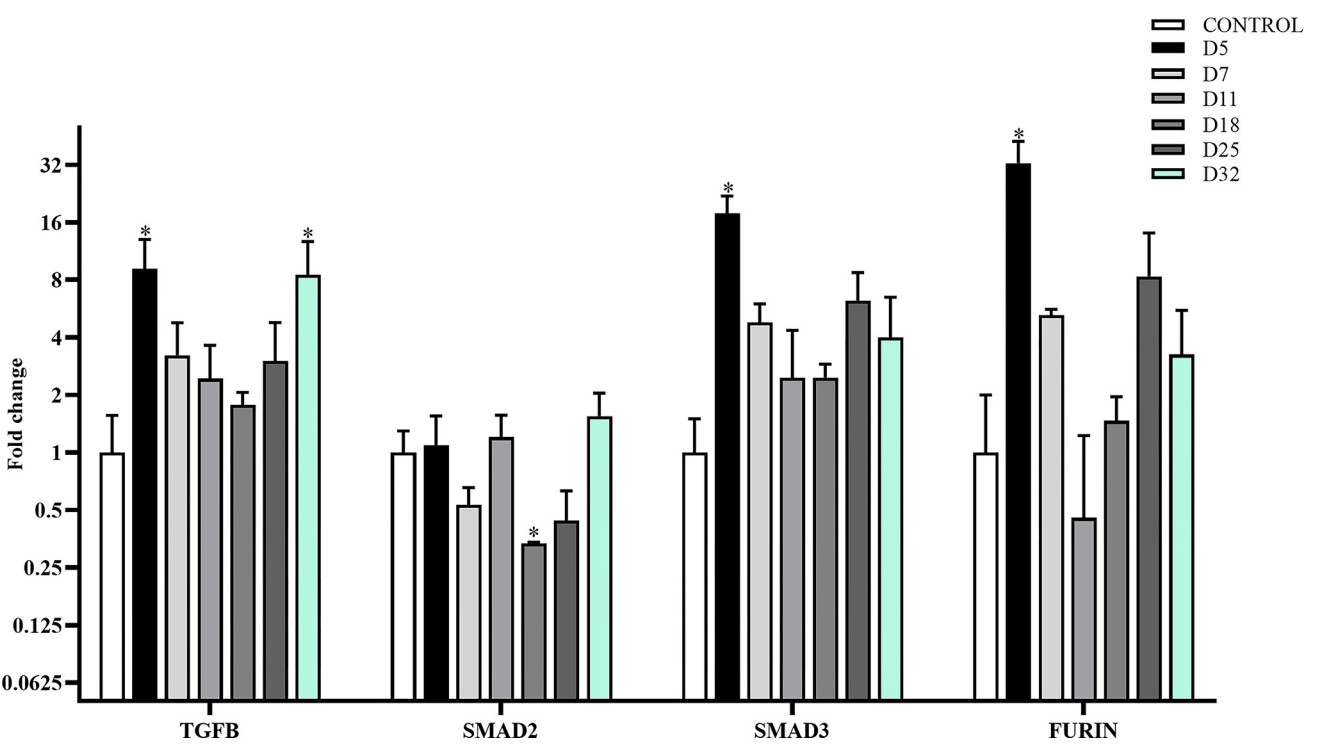

**Fig 3. Successful induction of cardiac fibrosis and up-regulation of pro-fibrotic and pro-inflammatory genes following the protocol #4.** Detection of mRNA levels of pro-fibrotic (COL1A1, α-SMA, TGF-ß, SMAD2, SMAD3, and FURIN) and pro-inflammatory (MMP9) genes through RT-qPCR in the mice treated with ISO (140 mg/kg) for 4 days, and then harvested on D5, D7, D11, D18, D25, and D32 days. Two waves of gene expression upregulation is evident for most of the tested genes, following the ISO treatment. Data are presented as mean ± SEM vs. control (n = 3). The mean expression is shown as a fold change. T Student's t-test: *p<0.05, **p<0.01, ***p<0.001.

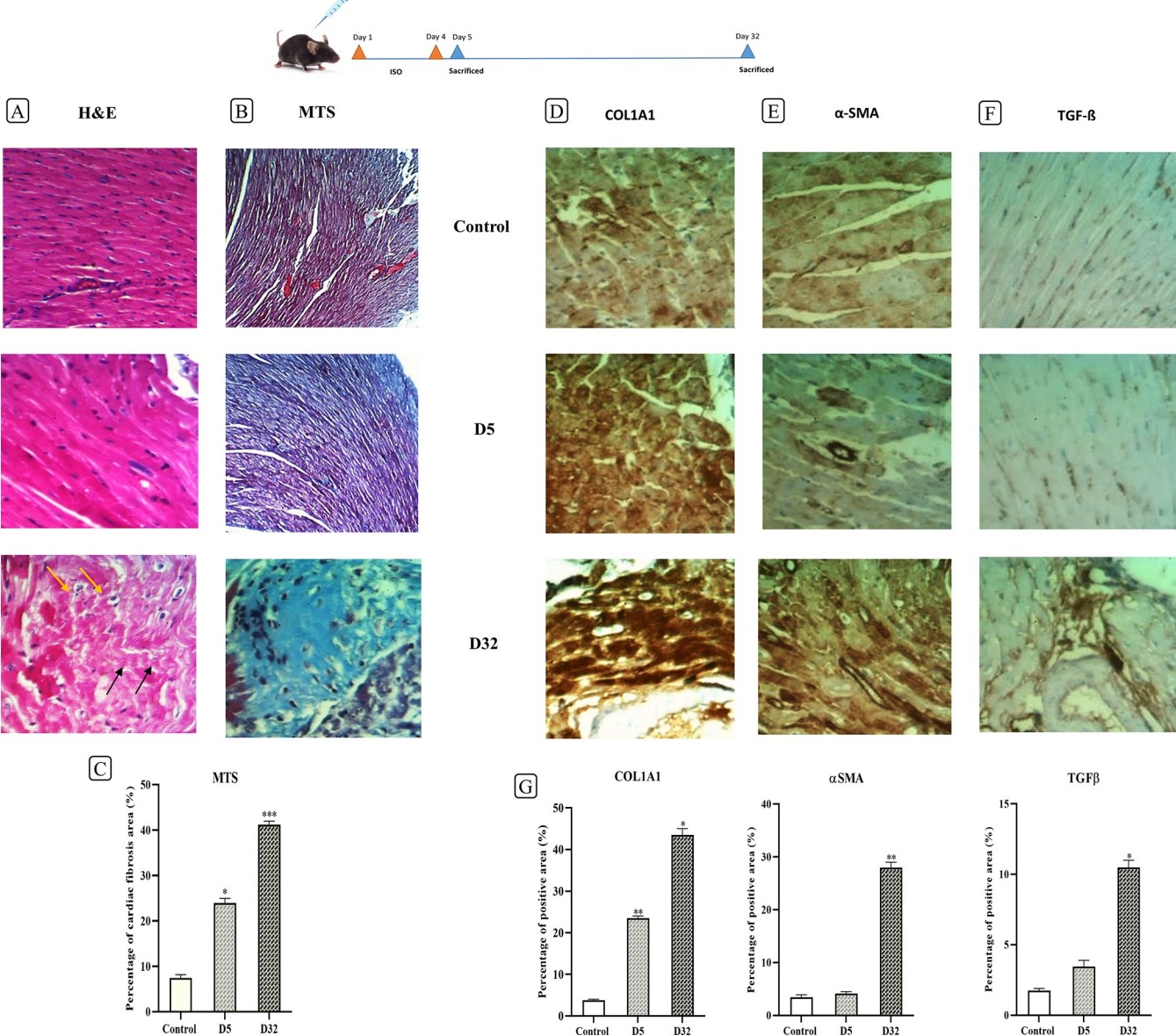

**Fig 4. Verification of the induced cardiac fibrosis through histopathological staining and measuring the marker proteins level following protocol #4. (A)** The H&E staining shows the morphological alteration of the cardiomyocytes in the left ventricular, following the ISO induction. Black arrows show the irregular organization of cardiomyocytes, and yellow arrows indicate the irregular shape of nuclei without any organelles in the cytoplasm. **(B)** MTS staining shows the amount of cardiac fibrosis induction (identified by blue-colored tissues). **(C)** Quantitative analysis of the fibrotic area extent through evaluating the collagen surface area in the left ventricle of mice in the D5 (*p<0.05) and D32 (***p<0.001) groups, in comparison to the control group. **(D-F)** show the Immunohistochemistry (IHC) staining of COL1A1 (D), α-SMA (E), and TGF-ß (F) in mice's left ventricular tissue sections on D5 and D32 after injection with ISO (140 mg/kg), compared to the control group. All these three marker proteins were upregulated/ accumulated on day 32nd, affirming the fibrosis induction. Nevertheless, COL1A1 (A) protein level has been drastically increased even on day 5th. Data are presented as the mean values ± SEM. Magnification (x400). H&E, hematoxylin and eosin; MTS, Masson's trichrome staining.

is blue, and cytoplasm is pink. On day 5, there is some formation of collagen fibers, but on day 32, the occurrence of cardiac fibrosis is fully evident in some loci. The percentage of collagen was calculated using color deconvolution and measurement of the threshold area for three groups (Fig 4C). The level of collagen on days 5 ($p<0.05$) and 32 ($p<0.001$), which is an indicator of fibrotic area, showed a significant increase compared to the control. Some sections of the whole heart tissue are shown in S2 Fig.

## Increased COL1A1, α-SMA, and TGF-ß protein levels following the cardiac fibrosis induction

To confirm the induction of cardiac fibrosis, immunohistochemical staining was carried out to investigate the alteration of COL1A1, α-SMA, and TGF-ß protein levels on D5 and D32, compared to the control group (Fig 4D–4F). The results (brownish area) were quantified using the Immunohistochemistry (IHC) Image Analysis Toolbox and the threshold area calculation in ImageJ (Fig 4G). This analysis verified that fibrosis induction through the novel protocol #4, has brought about the significant elevation of COL1A1 on both D5 ($P<0.01$) and D32 ($P<0.05$). However, α-SMA ($P<0.01$) and TGF-ß ($P<0.05$) showed a markedly increased protein expression level, only on D32 in comparison to the control group. Some sections of the whole heart tissue are shown in S2 Fig.

## Decreased expression of profibrotic genes following the DNC treatment in a mouse model of cardiac fibrosis

In the first group sacrificed on D5 (DNC/ISO+DNC), a significant decrease in COL1A1 (p-value = 0.02), α-SMA (p-value = 0.03), and MMP9 (p-value = 0.04) genes, the most important fibrotic markers, was observed compared to the mouse model of cardiac fibrosis on D5.

Related to the mouse sacrificed on D32, the expression of α-SMA, TGFβ and MMP9 genes was examined since only these genes remained up-regulated in the mouse model of cardiac fibrosis on D32. In the second group sacrificed on D32 (DNC/ISO+DNC), α-SMA (p-value = 0.0005), TGFβ (p-value = 0.02) and MMP9 (p-value = 0.004) genes were significantly decreased in comparison to the cardiac fibrosis. The administration of DNC for seven days following the ISO led to the significant down-regulation of α-SMA (p-value = 0.0002), TGFβ (p-value = 0.01) and MMP9 (p-value = 0.001) genes in the third group (ISO-DNC/D32) compared to the cardiac fibrosis (D32). In the fourth group (ISO-DNC/D25-D32), only the α-SMA gene showed a significant decrease (p-value = 0.002) (Fig 5A and 5B).

## Induction of pulmonary fibrosis through intraperitoneal ISO injection

Following the modified protocol #4, ISO was intraperitoneal-injected (140 mg/kg) into mice for 4 subsequent days. The reason for choosing intraperitoneal over subcutaneous injection is presented in (S1 File). Then the expression levels of pro-fibrotic and pro-inflammatory marker genes in the lung tissues were determined through RT-qPCR on days 5, 11, 18, 25, and 32. Among the genes of the TGF-ß signaling pathway, only TGF-ß showed a significant elevated expression level in D32. Nevertheless, FURIN expression level was also prominently, but not significantly, elevated (Fig 6). Non-TGFB marker genes were also upregulated in a sporadic pattern following the ISO injection. For example, COL1A1 expression level was increased on D5 and D11. The α-SMA expression level was increased on D11, D18, and D25. The COl3A1 expression level was increased on D11 and D18. MMP9 had increased expression levels both in D5 & D32.

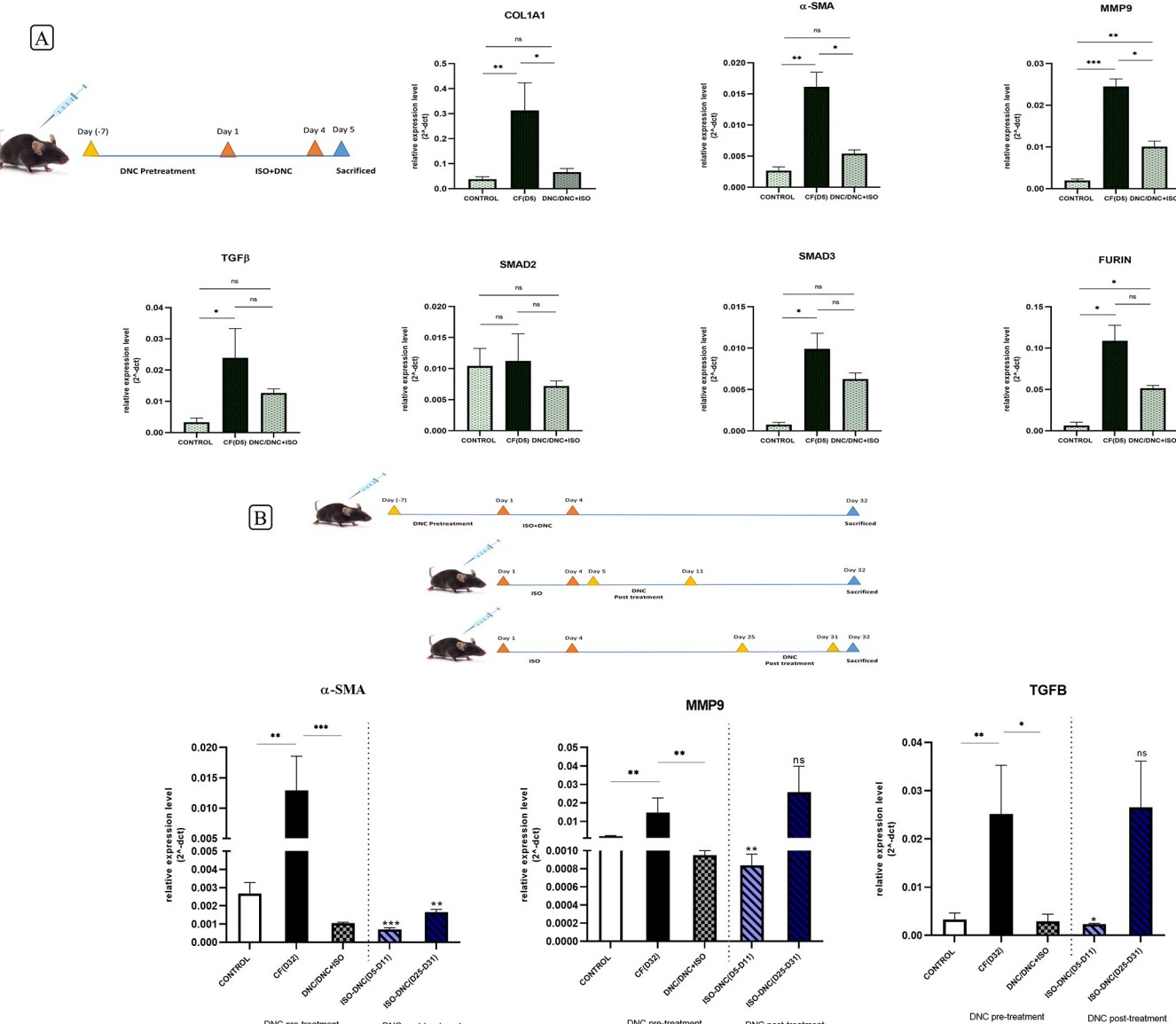

**Fig 5. DNC modulates the expression of genes related to cardiac fibrosis.** The expression of profibrotic and pro-inflammatory genes was determined by RT-qPCR following the DNC (10 mg/kg) administration. In the first and second groups sacrificed on (A) D5 and (B) D32, the mice were pre-treated with DNC for 7 days, followed by co-treatment of ISO and DNC for 4 days (DNC/ISO+DNC). In the third group, the mice were treated with ISO for 4 days, followed by DNC post treatment for 7 days and sacrificed on D32 (B). In the fourth group, the mouse was treated with ISO for 4 days and then from D25-D31 injected with DNC and sacrificed on D32 (B). Marker genes expression results indicated that pretreatment of mice with DNC, reduces the risk of cardiac fibrosis following the ISO treatment. However, after the fibrosis induction, DNC treatment is less effective, up to the delay of DNC treatment. Data are presented as mean ± SEM. The mean expression is shown as a relative expression level. One-way ANOVA test: ns: non-significance, *$p<0.05$, **$p<0.01$, ***$p<0.001$.

## Identification of DEGs

As we utilized samples from various studies that shared only cardiac fibrosis as a common factor, with different induction methods and species under investigation, all samples were not analyzed together or pooled. Initially, we analyzed the samples from each project separately and obtained the DEGs related to those samples.

DEGs were identified by analysis of raw read sequence using Galaxy. The DEGs of 4 GEO datasets have been reported separately in Table 2.

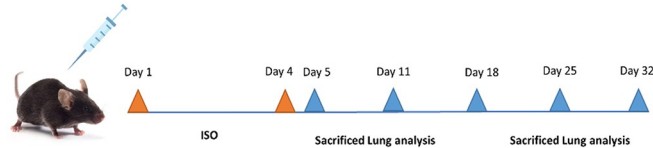

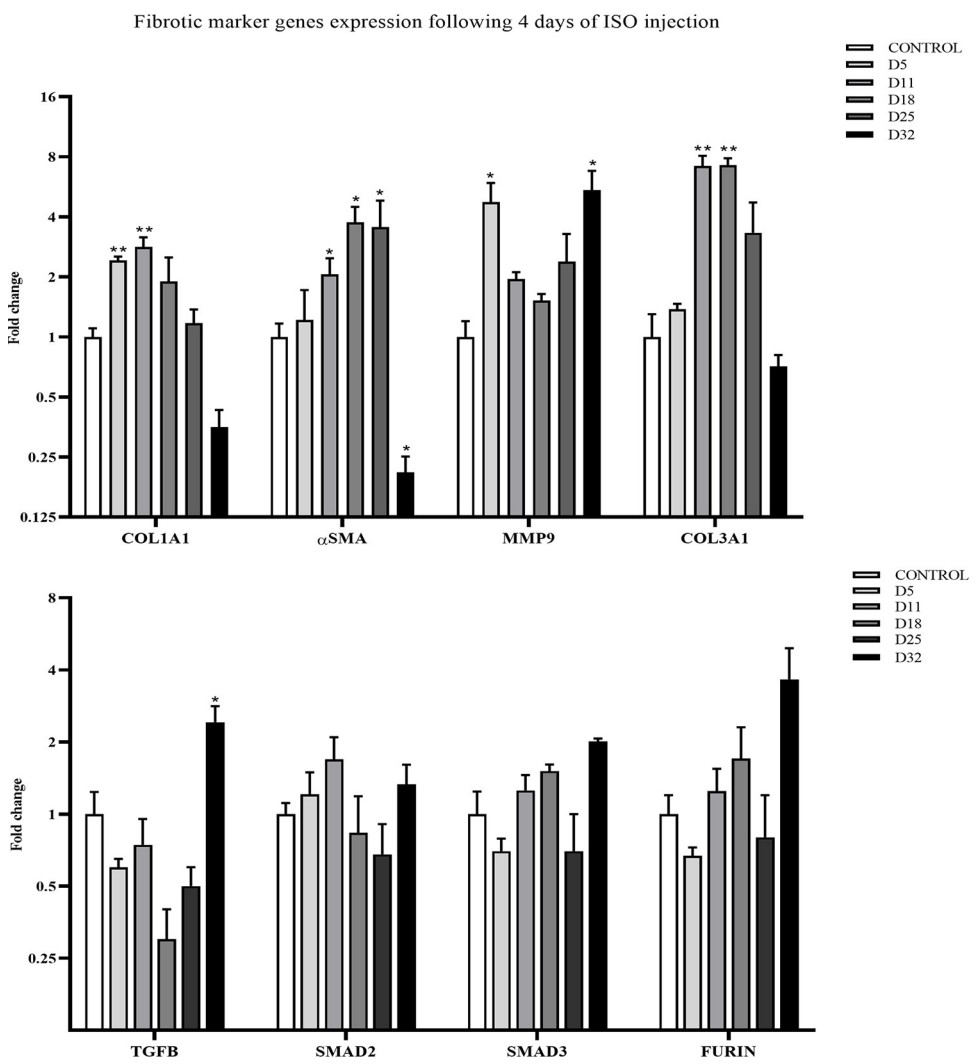

**Fig 6. Induction of pulmonary fibrosis based on the expression pattern of pro-fibrotic and pro-inflammatory genes.** RT-qPCR expression detection of pro-fibrotic (COL1A1, α-SMA, COL3A1, TGF-ß, SMAD2, SMAD3, and FURIN) and pro-inflammatory (MMP9) genes in the lung tissue samples prepared following the intraperitoneal-injection of 140 mg/kg ISO-chemical for 4 days and harvesting on D5, D7, D11, D18, D25, and D32. Data are presented as mean ± SEM vs. control (n = 3). The mean expression was shown as a fold change. T Student's t-test: *p<0.05, **p<0.01.

## Shared DEGs and enrichment analysis

To achieve our stated goal of finding a gene common to all these methods, we ultimately employed Venn analysis. By creating a Venn diagram, we were able to identify genes shared across all these datasets.

**Table 2. DEGs obtained from the RNA-seq data analysis.**

| GEO accession number | Organism/Tissue type | Samples | Up-regulated genes | Down-regulated genes |
|---|---|---|---|---|
| GSE97358 | Primary human cardiac fibroblast | 84 controls, 84 TGFB1-induced cardiac fibrosis at 24h | 268 | 286 |
| GSE123018 | Primary human cardiac fibroblast | 4 controls, 4 TGFB1-induced cardiac fibrosis at 24h | 126 | 191 |
| GSE152250 | Human cardiac fibroblast | 6 controls, 6 TGFB1-induced cardiac fibrosis at 24h and 48h | 2658 | 3827 |
| GSE151834 | Adult C57BL/6 mice/ fibrotic scar tissue | 40 adults C57BL/6 mice subjected to ischemic cardiac injury/ 20 samples injured region, 20 samples un-injured region | 9014 | 4119 |

After obtaining the DEGs in each project independently and dividing them into up-regulated and down-regulated, the Venn diagram was calculated. The Venn diagram shows 19 overlapping up-regulated and 10 overlapping down-regulated genes among these 4 datasets as illustrated in Fig 7A and 7B. To provide a comprehensive list of these DEGs, we have also

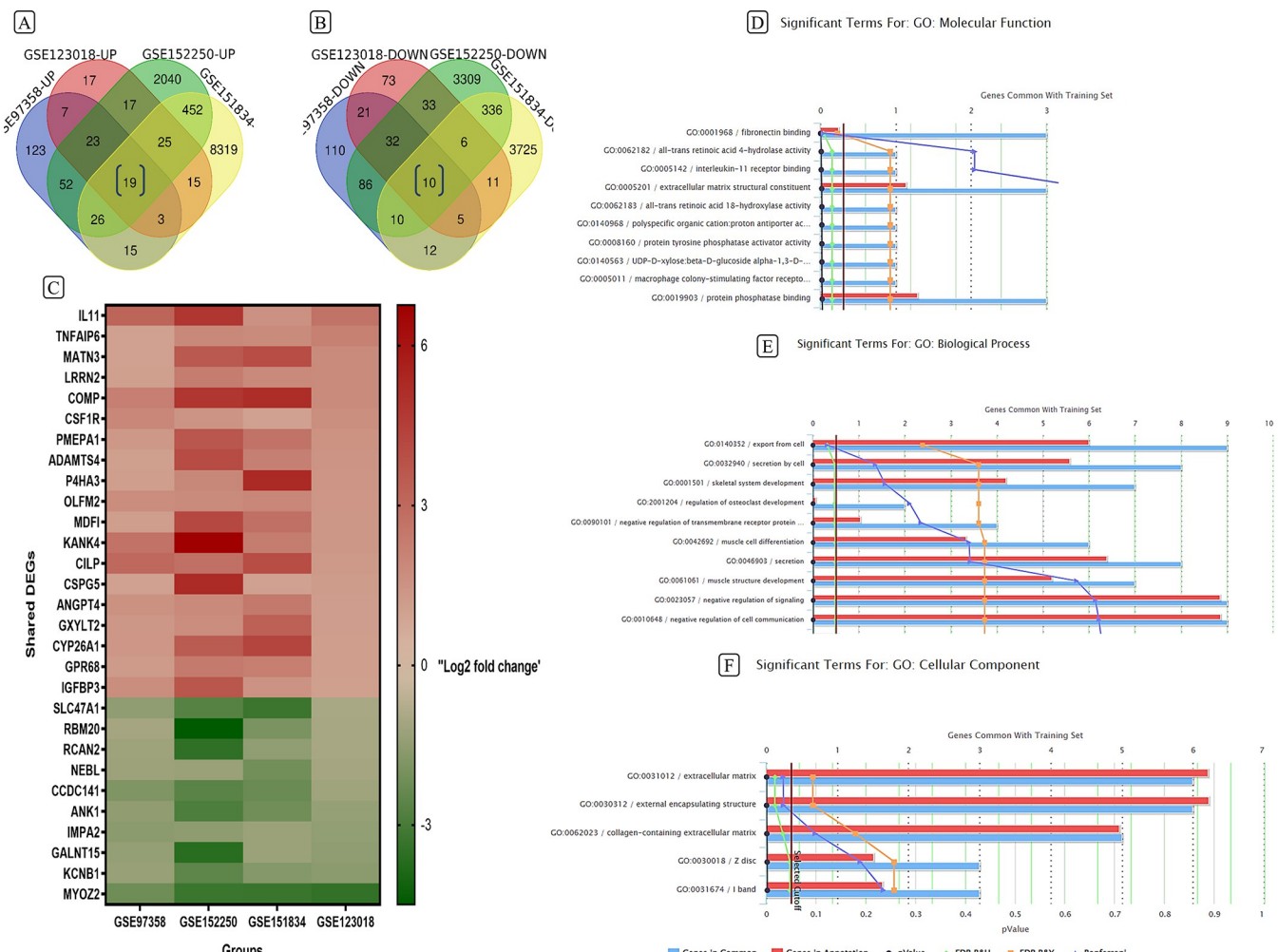

**Fig 7. Shared DEGs between the 4 GEO datasets and enrichment analysis.** (A) 19 shared up-regulated and (B) 10 shared down-regulated genes were calculated with a Venn diagram. (C) Heat map of 29 shared genes according to their Log2 fold change in each project. (D) Molecular function, (E) biological process and (F) cellular component of shared genes.

created a heatmap according to the Log2 fold change in Fig 7C. Additionally, we reported their molecular, biological, and cellular roles using functional enrichment analysis. The top 10 molecular functions and biological processes have been shown in Fig 7D–7F. Enrichment analysis of fibrosis-related genes reveals significant involvement of molecular functions associated with extracellular matrix interactions (fibronectin binding, ECM structural constituents) and signaling pathways (interleukin-11 receptor binding, protein phosphatase activities). Additionally, the presence of retinoic acid metabolism functions suggests potential roles for vitamin A derivatives in fibrotic processes. Regarding the biological process significant involvement of cell communication and signaling regulation processes, particularly negative regulation has been shown. It has also reported the major role of extracellular matrix production and secretion which is consistent with the characteristic excessive ECM deposition in fibrosis. Fig 7F displays the top cellular components relating to the association with extracellular matrix components, particularly collagen-containing structures. This underscores the central role of ECM remodeling in fibrotic processes, with potential implications for muscle tissue architecture.

## Protein-protein interaction network and hub genes analysis

29 common DEGs of the previous stage have been utilized for drawing the PPI network in both human (Fig 8A) and mice (Fig 8B). String is an online database and tool that collects and presents information about PPIs. It provides a combined score for each interaction. This score indicates the level of confidence in the existence of that interaction based on available evidence.

Cytoscape is an open-source software for visualizing molecular networks and biological pathways. It has the capability to import output data from STRING. By analyzing this network, genes with the highest number of connections to other genes (high degree in the network) are identified as hub genes. The interaction scores play a crucial role in this analysis, as interactions with higher scores are given more importance in identifying hub genes. Using the Cyto-Hubba plugin in Cytoscape, cartilage oligomeric matrix protein (COMP) was identified as the first hub gene in the top 10 groups for both human and mice with a score of 3. The other genes were MYOZ2, ADAMTS4, RBM20, NEBL, MATN3, ANGPT4, CSF1R and CILP. The COMP gene was chosen for examining expression changes in the present study.

## Validation of COMP as a hub gene in vitro and in vivo

Gene expression of COMP was measured in vitro and in vivo through qPCR following ISO-induced cardiac fibrosis and DNC treatment (Fig 8). An increase in the expression of this gene (p-value = 0.01) was observed following the use of ISO in cardiac fibroblasts and a significant decrease in its expression was reported (p-value = 0.03) following the co-treatment of ISO with DNC (Fig 8E). In the mouse model of cardiac fibrosis, there was a notable increase in gene expression on days 5, 18, and 25 (Fig 8F). Following the administration of DNC to the first group of mice (DNC/DNC+ISO-D5), the expression of the COMP gene decreased compared to the cardiac fibrosis D5 group. However, the decrease was not significant (Fig 8G).

## Discussion

In the current study, we demonstrated for the first time the effect of DNC on inhibiting cardiac fibroblasts activated by ISO, as well as its impact on a mouse model of cardiac fibrosis. We also observed an increase in the expression of genes involved in cardiac fibroblasts cultured on common polystyrene plates compared to those cultured on softwell plates without any treatment. Additionally, we introduced a new protocol for inducing cardiac fibrosis in mice using

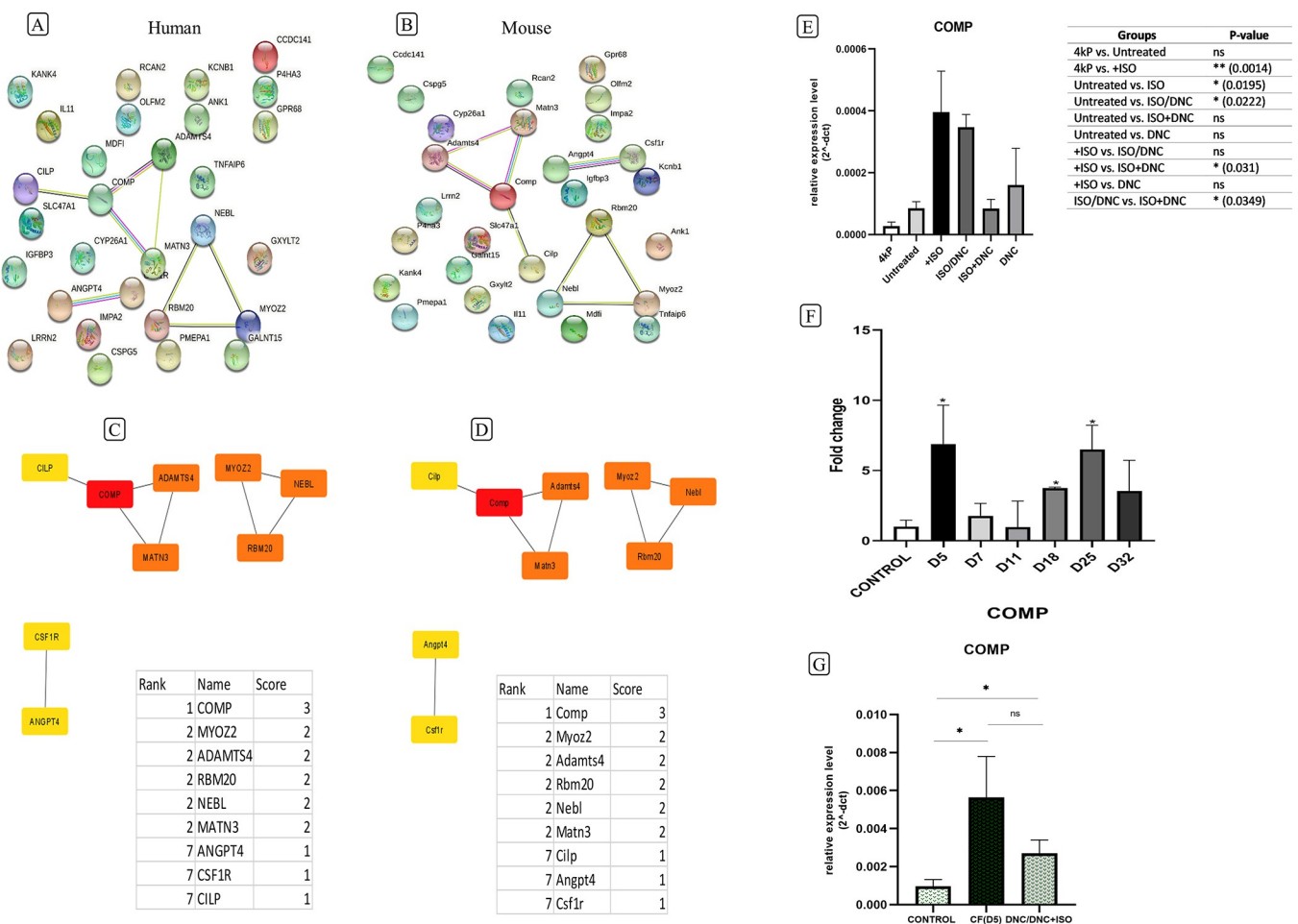

**Fig 8. Identifying the COMP gene as the first hub gene and confirming its increased expression in cardiac fibrosis.** PPI network of shared genes in (A) human and (B) mouse constructed by string. The top 10 hub genes for both (C) human and (D) mice; the COMP gene is the first one with a score of 3. (E) The expression of the COMP gene in cardiac fibroblast following ISO and DNC treatment with RT-qPCR. (F) The expression of the COMP gene in a mouse model of cardiac fibrosis and after the (G) DNC administration by RT-qPCR. Data are presented as mean ± SEM (n = 3). The mean expression is shown as the relative expression level and fold change. T Student's t-test, One-way ANOVA test: ns: non-significance, *p<0.05, **p<0.0.

ISO, as there was no standard dose for this purpose. Furthermore, we utilized ISO for the first time to induce pulmonary fibrosis. Finally, using a bioinformatics study the COMP gene was identified as a hub gene involved in cardiac fibrosis. To confirm this, we measured its expression level both in vitro and in vivo.

Myocardial fibrosis, a primary driver of ventricular remodeling, arises from the pathological restructuring of the extracellular matrix, culminating in fibrotic scarring, and can result from MI, manifesting as a chronic and progressive condition [41]. In other diseases, such as systemic hypertension, as well as diabetes, obesity and ageing, it causes interstitial fibrosis in the myocardium and subsequently results in reduced cardiac contractions and an increased risk of heart failure [5]. There are various available methods to induce cardiac fibrosis in animal models. These methods encompass genetic modifications, pharmacological interventions, surgical procedures, metabolic diseases, and dietary regimens [42–44]. The utilization of pharmaceutical compounds such as ISO-chemical represents a non-invasive approach. ISO, a non-selective β-adrenoceptor agonist [45, 46], causes fibrosis and cardiac hypertrophy, which helps study various aspects of cardiac dysfunction [47]. The administration of ISO-chemical can

cause significant stress on the heart muscle, activating the adrenergic system and leading to the expression of fibrotic factors. This process can ultimately result in cardiac remodeling and dysfunction [35, 48]. Curcumin has strong beneficial effects such as anti-inflammatory, antioxidant and cardioprotective potential [49]. Nano-curcumin, a polymer synthesized using dendrosomes, not only increases biostability but also enhances its solubility and intracellular absorption. DNC, a curcumin-loaded dendrosome nanocarrier, has demonstrated high efficacy in drug delivery across various studies. It was developed at Tarbiat Modares University [27, 50].

In this study, primary cardiac fibroblasts cultured on polystyrene plates showed a significant increase in the expression of COL1A1, SMAD2, TGFβ and FURIN genes compared to the softwell plates (4 kPa). Young's modulus is a mechanical property that measures elasticity. When fibrosis occurs, tissue stiffness increases (>100 kPa) [1]. The modulus of healthy heart tissue is about 2 kPa. The impact of stiffness has been studied using various hydrogels under cellular conditions. Due to this mechanical pressure, fibroblasts differentiate into myofibroblasts. Meanwhile, this conversion starts from the early hours of fibroblast isolation [51, 52]. Based on Natalie M. Landry et all's study, factors such as high passaging, substrate stiffness, high nutrition, and serum can affect fibroblast activation [23].

In the current study, the cells treated with 10 μm ISO for 24 hours showed an increase in the expression of fibrosis-related markers compared to the untreated group as well as the cells cultured on the softwell plate. After examining the cell cycle through flow cytometry, it was observed that the number of cells in the S phase increased while the number of cells in the G phase decreased compared to the control group, indicating the proliferation of fibroblasts and their activation. ISO induces the transition of the cell from the G1 phase to the S phase, a process regulated by various proteins. Among these proteins, CDKs play a crucial role as they promote cell cycle progression and proliferation by phosphorylating proteins such as cyclins when they are activated [53]. Following the use of DNC, the expression of profibrotic and pro-inflammatory genes decreased. However, co-treatment of ISO and DNC showed an even greater reduction effect. Additionally, there was an increase in the number of G1-phase cells, while the number of S-phase cells decreased, bringing the cells closer to their normal state. A study by MA Jin et al. demonstrated that curcumin reduces the proliferation of cardiac fibroblasts activated by angiotensin II and also inhibits cardiac fibrosis induced in rats. Curcumin was dissolved in water and administered via gavage for 28 days [49]. In our study, the cells treated only with DNC showed no significant difference from the untreated group, suggesting that curcumin can inhibit cells activated through the TGFβ pathway [54].

We [30] and others [33–37] have already performed sporadically successful protocols for the induction of cardiac fibrosis that caused us frustration. Looking for a reliable/repeatable protocol and given the absence of a standard dosage of ISO-chemical for inducing cardiac fibrosis, here, we introduced a novel protocol for fibrosis induction. In the current protocol, the expression of pro-fibrotic and inflammatory genes was examined at various time points following the ISO injection adding to the repeatability of the protocol. Cardiac fibrosis includes three stages: the initial phase, the effective phase and the proliferation phase. Fibrosis initiates with increased expression of cytokines and pro-fibrotic growth factors such as TGF-β after stimulation. With the binding of these cytokines and growth factors to their receptors, an effective response begins and ultimately leads to the proliferation of differentiated myofibroblasts, which includes elevated expression levels of α-SMA and MMPs [12, 20, 55].

There was a report of injecting 170 mg/kg of ISO-chemical for the induction of cardiac hypertrophy [38]. However, the application of such a dose of ISO-chemical resulted in drastic lethality of the animals. This event was attributed to the quality of the newly provided batch of chemicals which is light sensitive [56]. Meanwhile, subcutaneous injection of ISO at a dose of

140 mg/kg for 4 days brought about widespread changes in the expression of fibrosis-related marker genes, with an ignorable lethality rate in treated mice.

Accordingly, SMAD3, FURIN, and COL1A1 genes were upregulated at the early stages of the differentiation process as an induction phase. FURIN was the main early marker gene to be upregulated (25x) more than the other early marker genes (~16x & 10x). Interestingly, only COL1A1 was also upregulated at the early stages of pulmonary fibrosis. The RT-qPCR results suggested the COL1A1 as a common early marker gene for fibrosis incidence in both organs. Consistently, the accumulation of COL1A1 protein during the early stage of cardiac fibrosis incidence was verified through IHC analysis. Overall data suggest COL1A1 as the main and common early marker gene for the incidence of cardiac and pulmonary fibrosis.

Unlike the early markers, the expression level of (α-SMA, TGF-ß, and MMP9) was changed both at the early and late stages of cardiac fibrosis (induction and stabilization phases). Accordingly, α-SMA, TGF-ß, and MMP9 were upregulated like the literature reports [12, 57]. Due to the variation, COL3A1 expression level in cardiac fibrosis is not shown while its expression level was increased in the fibrotic lung tissues samples of the same animals.

The extracellular matrix (ECM) in healthy adult human hearts is primarily made up of fibrillar collagen types I (COL1) and III (COL3). These two collagen types have different molecular structures, with COL1 fibrils stiffer than COL3 [58]. Cardiac fibrosis is characterized by increased ECM deposition, and collagen cross-linking [59]. Various heart conditions like atrial fibrillation result in an elevation of the COL1/COL3 ratio [60, 61]. Increased COL1A1 and its expression level in the affected heart tissue samples of our experiment are in accordance with these reports.

MMPs (Matrix Metalloproteinases) are enzymes that play a crucial role in collagen degradation and cardiac remodeling [62]. MMP-9 significantly increases cardiovascular diseases such as hypertension, atherosclerosis, fibrosis, and myocardial infarction. The abundance of literature on MMP-9 highlights its importance as a promising biomarker [63]. This is consistent with the increased level of MMP9 following the cardiac fibrosis induction.

Fibrosis is an extremely complex process. Moreover, isoproterenol is a chemical agent that acts broadly and induces multiple types of cardiac fibrosis (replacement, interstitial, and perivascular) [33, 64, 65]. Specific signaling pathways regulate each of these types, potentially creating a complex network of interactions [66]. Consequently, during the intermediate days (D7, 11, 18, 25) when some genes show no change, they may be influenced by these diverse pathways. It's possible that during these days, genes outside the TGF-β pathway might exhibit significant expression changes.

We conducted histopathological examinations like H&E and MTS to investigate the histological tissue changes for control, D5, and D32 mice groups. Tissues from days 5 and 32 were chosen for staining due to the significant gene expression changes we detected during our RT-qPCR analysis. IHC was also performed on these three groups to investigate COL1A1, α-SMA, and TGF-β protein expression. In H&E staining, cardiomyocyte degeneration was not observed in the control and D5 groups. However, on D32, degeneration and necrosis were observed in cardiomyocytes. In MTS, there was no evidence of cardiac fibrosis in the control group, and collagenous areas were only positive in their regular positions (such as vascular walls). However, a density of immature collagen fibers was forming on D5. Consistently, fibrosis with a moderate to severe degree was detected in some foci on D32. The level of reactivity in α-SMA and TGF-β was normal and primarily positive in vascular walls in the control and D5 groups. However, their response was feeble in cardiomyocytes. While the level of COL1A1 reactivity in cardiomyocytes was deficient in the control group, its level was increased in a scattered pattern on D5, with a low to moderate degree. On D32, α-SMA level was increased in cardiomyocytes, indicating a myocardial injury trend. TGF-β and COL1A1 were also rising in

cardiomyocytes. The initial upregulation of gene expression often rapidly responds to injury or stress, while the accumulation of protein and corresponding histological changes take longer to manifest. Proteins such as collagen have a longer half-life compared to RNA, thus even after the gene expression levels decline, the proteins can continue to accumulate and persist in the tissue [20, 66–68]. Therefore, in our study, the expression level of collagen had returned to its normal state by D32, while the collagen fibers had matured histologically. Moreover, its protein expression had increased compared to D5. Proteins can have much longer half-lives than mRNA, allowing them to accumulate even when transcription and translation have slowed [69]. Thus, according to the protein levels, the D32 group represented the mice with occurrence of the cardiac fibrosis. In the study by Kuwahara et al., the protein expression of TGF-β was observed in the pressure overload (PO) model from the third day onwards until the 28[th] day, and there was no expression in the early days [70].

Bleomycin has long been used as an antibiotic and chemotherapy compound to introduce a pulmonary fibrosis model. The most common way to administrate bleomycin is intratracheal, but due to the surgical incision at the level of the trachea, it is associated with significant mortality during the operation [71, 72]. Here, for the first time, we aimed to use ISO-chemical to introduce a less lethal pulmonary fibrosis method along with cardiac fibrosis induction, without any need for surgery. At first, we used the subcutaneous method of ISO injection, which ended in no molecular marker expression alteration in lung tissues (S1 File). Then, the intra-peritoneal injection method was applied using protocol#4 and some of the fibrosis-related marker genes (COL1A1 (3x), α-SMA (4x), TGF-ß (2x), COL3A1 (8x), and MMP9 (6x)) were upregulated. Although all of the fibrosis-related marker gene expressions were altered during the cardiac fibrosis induction, only COL1A1 and MMP9 expression elevation was observed on D5 of the process. These RT-qPCR-detected expression elevations of COL1A1 and MMP9 in the lung tissue samples of our experiment are by the reported COL1A1 [73] and MMP9 [74] expression elevation reported by others. Also, increased expression level of α-SMA, on days 11, 18 and 25 of the process is in accordance to the other reports [75].

The selection of DNC injections as a treatment modality was made for clinical purposes. Pre-treatment and co-treatment with DNC are employed to assess their effectiveness in preventing cardiac fibrosis. Furthermore, post-induction DNC treatment aims to explore its potential inhibitory effects on fibrosis progression and its therapeutic utility. In the prevention group, notable reductions in the expression levels of COL1A1, α-SMA, and MMP9 were observed in mice sacrificed on Day 5, compared to those in the cardiac fibrosis group. Additionally, in mice exhibiting fibrosis on Day 32, wherein three markers (α-SMA, TGFβ, MMP9) were elevated, administration of DNC led to significant decreases in their expression levels. In the second group, with the aim of inhibition and therapy, a decrease in the expression of the three mentioned markers was observed on D32. However, when DNC was administered from D25 to D31, this reduction in expression was observed only in α-SMA. post-treatment with DNC in the final days, specifically from day 25 to 31, cannot be very effective in reducing the expression of genes involved in fibrosis. DNC post-treatment should be administered immediately following isoproterenol use to be effective. Anti-inflammatory effects of curcumin during the acute phase, are demonstrated by enhanced macrophage apoptosis and reduced secretion of key pro-inflammatory cytokines like IL-6, IL1b, and TNF-α. This modulation of macrophage activity also influences resident cardiac fibrosis, leading to diminished IL-18 expression in fibroblasts, impaired phosphorylation of SMAD2/3 in cardiac fibroblasts, decreased collagen synthesis, and ultimately, the preservation of long-term cardiac function post-myocardial infarction [24]. In atrial fibrillation (AF), inflammation and myocardial fibrosis play key roles in its development. Curcumin has been found to effectively counteract myocardial fibrosis and inflammation, potentially offering therapeutic benefits for AF. This protective effect of

curcumin may be linked to its impact on the IL-17 signaling pathway. Additionally, pivotal genes like COL1A1, FANS, PCK1, BMP10, IL33, and FIGF are implicated in curcumin's mechanisms of action against AF [76].

In the present study, using raw RNA-seq data analysis (GSE97358, GSE123018, GSE152250, GSE151834), we have found the COMP gene as a hub gene that could be associated with cardiac fibrosis. COMP is one of the ECM proteins primarily studied in the context of tendons and cartilage. The main function of COMP is direct interaction with other ECM components, including collagens and TGF-β1, and facilitating the stability of the ECM network by forming collagen fibers. This role is crucial for maintaining cardiac homeostasis. Studies have shown that COMP expression is increased in liver fibrosis, and also cardiac fibrosis [77]. Prior research has established that abnormal COMP expression leads to myocardial cell apoptosis and loss of myofilament integrity. Additionally, COMP, along with COL1A1, COL1A2, and PRELP, exhibit upregulation in ECM-receptor interaction and focal adhesion pathways, consequently inducing alterations in the extracellular matrix composition [28]. Our investigation revealed a noteworthy upregulation of COMP gene expression in cardiac fibroblasts induced with ISO compared to the control group. This elevation in COMP gene expression underscores its potential implication in the fibrotic process. Moreover, the administration of DNC resulted in a subsequent decrease in COMP gene expression. Furthermore, employing a mouse model of fibrosis, we observed a significant increase in COMP gene expression on days 5, 18, and 25 post-induction. These findings underscore the dynamic nature of COMP gene expression during the progression of cardiac fibrosis, implicating its involvement in the fibrotic cascade.

In conclusion, our study demonstrated that cultivating cardiac fibroblasts on polystyrene plates induced mechanical activation, which can be reduced by using softwell plates. Additionally, the mouse model of cardiac fibrosis (140 mg/kg for 4 days) utilized here exhibited significant alterations in the expression of cardiac fibrosis markers with minimal lethality. Also, using this protocol but with a different type of injection, the expression changes of genes related to pulmonary fibrosis can also be observed. DNC, acting through the TGFB pathway, effectively reduced the activation and proliferation of cardiac fibroblasts. Furthermore, it moderated cardiac fibrosis by downregulating pro-fibrotic and pro-inflammatory gene expression. Lastly, the COMP gene was identified as a potential marker associated with cardiac fibrosis.

## Supporting information

**S1 Table. Primer sequences used for RT-qPCR analysis.**
(DOCX)

**S1 Fig. Induction of cardiac fibrosis wasn't observed based on previous protocols.** (A) The expression of COL1A1 and α-SMA genes was evaluated through RT-qPCR after ISO injection (10 mg/kg for 3 days and 5 mg/kg for 11 days) with mice being harvested on day 15 and day 25 compared to the control group. (B) The mRNA level of COL1A1 and α-SMA genes in ISO (5 mg/kg and 10 mg/kg for 11 days) treated mice was determined by RT-qPCR compared to the control group. Data are presented as mean ± SEM vs control (n = 3). The mean expression was shown as a fold change. T Student's t-test: non-significance.
(TIF)

**S2 Fig. The entire heart tissue sections.** Complete images of heart tissue for MTS and IHC for markers COL1A1, α-SMA, and TGF-β were taken using a 4x microscope lens.
(TIF)

**S1 File. Choosing the injection type to induce pulmonary fibrosis.**
(DOCX)

## Acknowledgments

We would like to thank Mr. Amin Baninajar for his technical expertise and assistance.

## Author Contributions

**Conceptualization:** Behnaz Beikzadeh, Majid Sadeghizadeh, Bahram M. Soltani.

**Formal analysis:** Behnaz Beikzadeh, Bahram M. Soltani.

**Investigation:** Behnaz Beikzadeh, Mona Khani, Yasamin Zarinehzadeh.

**Methodology:** Behnaz Beikzadeh, Mona Khani, Bahram M. Soltani.

**Resources:** Behnaz Beikzadeh, Elham Abedini Bakhshmand, Shahram Rabbani, Bahram M. Soltani.

**Software:** Behnaz Beikzadeh.

**Supervision:** Bahram M. Soltani.

**Validation:** Behnaz Beikzadeh, Mona Khani, Majid Sadeghizadeh, Bahram M. Soltani.

**Writing – original draft:** Behnaz Beikzadeh, Mona Khani, Yasamin Zarinehzadeh.

**Writing – review & editing:** Majid Sadeghizadeh, Shahram Rabbani, Bahram M. Soltani.

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
