## [Decision Letter · Decision Letter 0]

15 Jul 2024

PONE-D-24-22014Preventive and treatment efficiency of Dendrosomal Nano-Curcumin against ISO-induced cardiac fibrosis in mouse modelPLOS ONE

Dear Dr. Soltani,

Thank you for submitting your manuscript to PLOS ONE. After careful consideration, we feel that it has merit but does not fully meet PLOS ONE’s publication criteria as it currently stands. Therefore, we invite you to submit a revised version of the manuscript that addresses the points raised during the review process.

We look forward to receiving your revised manuscript.

Kind regards,

Rami Salim Najjar, Ph.D.

Academic Editor

PLOS ONE

Journal Requirements:

Additional Editor Comments:

Authors present an interesting study on the effects of nano curcumin iso-induced fibrosis. I have provided comments below:

**Major comments:**

Authors must provide data validating that they have fibroblasts and not VSMCs or other cell types.

Regarding histology: Authors should show the entire heart on the slide with XY stitching to confirm that the representative image reflects a similar degree of fibrosis between groups.

Authors should also take the images at the same time with the same color contrast and white background adjustments to ensure that the color is the same between images so that they can be adequately assessed.

**Minor comments:**

What was the vehicle that DNC was in for cell treatment?

For cells, how was ISO dose determined? Is there MTT data available for this compound?

Reviewers' comments:

Reviewer's Responses to Questions

**Comments to the Author**

1. Is the manuscript technically sound, and do the data support the conclusions?

Reviewer #1: No

2. Has the statistical analysis been performed appropriately and rigorously? 

Reviewer #1: I Don't Know

3. Have the authors made all data underlying the findings in their manuscript fully available?

Reviewer #1: Yes

4. Is the manuscript presented in an intelligible fashion and written in standard English?

Reviewer #1: Yes

5. Review Comments to the Author

Reviewer #1: Manuscript Number: PONE-D-24-22014

This manuscript by Beikzadeh et al. sought to optimize the concentration of isoproterenol (ISO)-induced cardiac fibrosis and to assess the beneficial effects of curcumin in ISO-induced animal model of fibrosis. The study was well performed, however, there are serious technical issues with the study as a whole.

1- The data presented clearly demonstrate that the fibrosis model (protocol #4; 140 mg/kg ISO) is unreliable, as the levels of fibrotic markers are inconsistently expressed when in fact they should be up-regulated if cardiac fibrosis is evident. Please explain the differential expression of these markers.

2- Can the authors explain/discuss the significance of their novel protocol in light of the different phases of fibrosis (inflammation, proliferation, and maturation phases)?

3- Data in Fig 4d,g contradicts the data in Fig 3a and shows more Col1a1 is present on D32 vs D5.

4- Data in Fig 5 are very confusing and lack clear rationale and interpretation. The apparent upregulation of TGFb, MMP9 with ISO-DNC is problematic and indicates non-specific effect of DNC.

5- Was there any evidence ISO triggers pulmonary fibrosis?

6- The comparison of the RNAseq data should be interpreted with caution as the analyses were performed on completely different samples, human fibroblasts stimulated with TGFb in vitro and mice hearts subject to LAD ligation. It would have been conceivable had the authors have used RNAseq generated from human ischemic hearts (which are widely available) vs mice LAD. Also problematic is the stimulation times, 24hs in human vs days/weeks in mice. In addition, the RNAseq analysis is poorly explained and presented.

7- How do the authors explain the fluctuation of COM expression in Fig8F?

6. PLOS authors have the option to publish the peer review history of their article (what does this mean?). If published, this will include your full peer review and any attached files.

Reviewer #1: No

---

## [Author Response · Author response to Decision Letter 0]

26 Aug 2024

Response to Reviewers

Submission ID: PONE-D-24-22014

Manuscript Title: Preventive and treatment efficiency of Dendrosomal Nano-Curcumin against ISO-induced cardiac fibrosis in mouse model

Dear Editor;

I would like to thank the reviewers for their careful reading of this manuscript and their suggestions and thoughtful comments which helped to improve the quality of this manuscript. 

Our response follows:

Reviewers' comment: Authors must provide data validating that they have fibroblasts and not VSMCs or other cell types.

Response: We have used a well-established protocol reported in the manuscript (reference # [30]), to isolate fibroblast from the mouse cardiac mixture of cells. Accordingly, α-SMA and also other marker genes (vimentin and fibronectin) expression are used to differentiate fibroblasts from the other cells in the preparation. Along with molecular markers, morphology of the cells were used to support the identity of the fibroblasts (1). Similarly, references to methods from other studies have also been included. As stated in the manuscript, α-SMA expression was measured and reported because a key distinctive feature differentiating myofibroblasts from fibroblasts and other cardiac cells is the presence of α-SMA. The expression of α-SMA is the gold standard for identifying activated fibroblasts (2). 

In details, cardiac fibroblast cell culture has been performed globally for many years, with established protocols. In the final stage, after 3 hours, fibroblasts adhere to the plate while other cells are removed. Numerous articles on cardiac fibrosis have been published using primary cardiac fibroblast cell cultures without any tests to confirm their fibroblast nature (3-5). However, despite repeating this procedure, we used molecular testing for confirmation. This content is placed in the section Primary Cardiac Fibroblast Culture (Page 4).

Reviewers' comment: Regarding histology: Authors should show the entire heart on the slide with XY stitching to confirm that the representative image reflects a similar degree of fibrosis between groups. Authors should also take the images at the same time with the same color contrast and white background adjustments to ensure that the color is the same between images so that they can be adequately assessed.

Response: Since the project has been accomplished, providing such figures means repeating all of the experiments. In order to address this issue, we supplemented our manuscript with images of some complete cardiac sections in the supplementary file (S2_Fig). 

For COL1A1 on day 32, where two images are placed, they complement each other. Due to the lack of a Microscope Scanner, two images were taken side by side to show the complete image of the tissue section with a 4x lens.

In Figure 4, where images with 400x zoom were placed, these images were taken by a pathologist from cardiomyocytes and areas in the left ventricle of the heart that are involved in fibrosis. For example, we naturally have collagen expression in parts of the heart such as vascular walls, or α-SMA and TGF-β show normal expression in vascular walls in the control group and on day 5. Therefore, the images in Figure 4 were taken from abnormal areas such as cardiomyocytes and evaluated using ImageJ. The interpretation and explanation of this section are provided in the discussion part.

Reviewers' comment: What was the vehicle that DNC was in for cell treatment?

Response: Dendrosome was used as a nanocarrier for curcumin. 

Nano-curcumin, a polymer synthesized using dendrosomes, not only increases biostability but also enhances its solubility and intracellular absorption. DNC, a curcumin-loaded dendrosome nanocarrier, has demonstrated high efficacy in drug delivery across various studies. It was developed at Tarbiat Modares University (Discussion, page 14). In the Materials and Methods section, under the MTT subsection, the purchase of this substance and its protocol are described, along with the provided reference.

Reviewers' comment: For cells, how was the ISO dose determined? Is there MTT data available for this compound?

Response: This dose ISO (10µm) has been widely used in various in vitro studies. Two references supporting the use of this dose have been added to the manuscript text (Material and methods - ISO and DNC treatment, page 5, line 146).

Reviewers' comment: The data presented clearly demonstrate that the fibrosis model (protocol #4; 140 mg/kg ISO) is unreliable, as the levels of fibrotic markers are inconsistently expressed when in fact they should be up-regulated if cardiac fibrosis is evident. Please explain the differential expression of these markers.

Response: If we just focus on the RT-qPCR results, there seems unexpected results of gene expression during the days 7th to 25th. However, looking at Fig.3 RT-qPCR analysis shows that TGFB, SMAD3, Furin, Col1A1, aSMA, MMP9 all have been upregulated in day 5th of the process, consistent to the MTS (Fig. 4). Also, TGFB, aSMA, and MMP9 expression are again upregulated in day 32 as the second wave of gene expression upregulation, consistent to the IHC (Fig. 4). As explained in the article, day 5 showed the beginning of collagen formation histologically, while on day 32, collagen fibers were visible in the left ventricle, and SMA changes were well-demonstrated through IHC. This indicates the persistence of fibrotic conditions even 28 days after the initiation of this process. We expected that on day 32, these changes would be more evident at the tissue level rather than at the RNA level, which was indeed the case.

Please take into the consideration that fibrosis is an extremely complex process and isoproterenol induces multiple types of cardiac fibrosis (replacement, interstitial, and perivascular)(6-8). Therefore, specific signaling pathways regulate each of these types, potentially creating a complex network of interactions and gene expression alterations(9). Consequently, during the intermediate days (D7, 11, 18, 25) when some genes show no change, they may be influenced by these diverse pathways. It's possible that during these days, genes outside the TGF-β pathway might exhibit significant expression changes. For instance, on day 32, TGF-β might have been activated through an alternative signaling pathway, which subsequently led to the observed increase in SMA expression. The crucial point is that we observed clear changes in these genes on day 5, indicating the induction of fibrosis. This observation is significant in demonstrating the onset of the fibrotic process. It is not clear why the upregulation of marker genes follows the double wave model of gene expression upregulation. Double wave of stress response is not unusual in different organisms including plant and animals.

There is no standardized dose for inducing fibrosis with ISO, and each study has used a different dose. However, a common feature among many studies is that they have used the day immediately following isoproterenol injection as the day to examine cardiac fibrosis. In our study, day 5 (one day after consecutive ISO injections) showed extensive expression changes in cardiac fibrosis markers. This indicates that day 5 is crucial for initiating the fibrosis process. 

(Discussion, Page 16, line 564-569)

Reviewers’ comment: Can the authors explain/discuss the significance of their novel protocol in light of the different phases of fibrosis (inflammation, proliferation, and maturation phases)?

Response: According to the time/tissue serials of RT-qPCR investigation and also intensity of the fibrotic-markers staining, we came to the conclusion that regardless of fibrosis type, there could be two phases of cardiac fibrosis induction (day 5) and stabilization (day 32). Nevertheless, since isoproterenol is a compound capable of inducing all three models of cardiac fibrosis, distinguishing them is hard and it is feasible when we specifically induce and examine a single model of it.

In the discussion section of the article, based on the results obtained from this model on the two main days (5 and 32), we have discussed the early and late expression of these markers. Some markers showed a significant increase in expression only in the early days as an induction phase, while others showed increases in both early and late (day 32) stages (induction and stabilization phases). This pattern has been reported for both cardiac and pulmonary fibrosis.

Reviewers’ comment: Data in Fig 4d,g contradicts the data in Fig 3a and shows more Col1a1 is present on D32 vs D5.

Response: The initial upregulation of gene expression often rapidly responds to injury or stress, while the accumulation of protein and corresponding histological changes take longer to manifest. Proteins such as collagen have a longer half-life compared to RNA, thus even after the gene expression levels decline, the proteins can continue to accumulate and persist in the tissue (9-12). Therefore, in our study, the expression level of collagen had returned to its normal state by day 32, while the collagen fibers had matured histologically. Generally speaking, proteins can have much longer half-lives than mRNA, allowing them to accumulate even when transcription and translation have slowed (13). 

(Discussion, Page 16, line 582-588)

An example from the study on Cx43 and NaV1.5 expression:

The observed reduction in Cx43 protein level correlated with downregulation of Gja1 RNA expression at postnatal weeks 1 and 2 in MHC-CnA. However, at weeks 3 and 4 the RNA level of Gja1 normalized to WT values, whereas the protein level remained reduced. This suggests that the initial Cx43 reduction is caused at the transcriptional level, whereas at weeks 3 and 4 another mechanism may be responsible (14). 

Reviewers’ comment: Data in Fig 5 are very confusing and lack clear rationale and interpretation. The apparent upregulation of TGFb, MMP9 with ISO-DNC is problematic and indicates non-specific effect of DNC.

Response: For DNC injection in mice, we had four groups, which are schematically represented in this figure for better understanding. 

The selection of DNC injections as a treatment modality was made for clinical purposes. Pre-treatment and co-treatment with DNC are employed to assess their effectiveness in preventing cardiac fibrosis. Furthermore, post-induction DNC treatment aims to explore its potential inhibitory effects on fibrosis progression and its therapeutic utility. Actually, we intended to mimic the heart attack (fibrosis induction) and DNC-treatment of it before and after the attack. Suppose that we have a drug called DNC and want to pre-treat or post- attack treating the patient with this drug. 

Part A, mimics someone who is prescribed with DNC for 7 days before the induction and during the fibrosis induction. Results showed that pretreatment with DNC for 7 days, followed by co-treatment of ISO and DNC for 4 days (DNC/ISO+DNC) drastically reduces fibrosis marker genes expression immediately after the fibrosis induction (day 5th).

Part B, also mimics someone who is prescribed with DNC for 7 days before the induction and during the fibrosis induction. However, it measures the effect of DNC pretreatment and cotreatment on fibrosis progression in long term (day 32). This type of DNC-pretreatment also attenuated the fibrotic marker genes expression in long term (shown by column DNC/DNC+ISO sign). 

If it is immediately applied after the heart attack (fibrosis induction), post-treatment of DNC is also effective for attenuating the fibrotic genes expression (shown by ISO-DNC (D5-D11) in the graphs). 

However, if DNC is applied with delay (day 25th) after the heart attack (fibrosis induction by ISO), its protective effect won’t be reliable. 

In another word, in the prevention group, notable reductions in the expression levels of COL1A1, α-SMA, and MMP9 were observed in mice sacrificed on Day 5, compared to those in the cardiac fibrosis group. Additionally, in mice exhibiting fibrosis on Day 32, wherein three markers (α-SMA, TGFβ, MMP9) were elevated, administration of DNC led to significant decreases in their expression levels.

In the second group, with the aim of inhibition and therapy, a decrease in the expression of the three mentioned markers was observed on D32. However, when DNC was administered from D25 to D31, this reduction in expression was observed only in α-SMA. This indicates that post-treatment with DNC in the final days, specifically from day 25 to 31, cannot be very effective in reducing the expression of genes involved in fibrosis. The two markers, TGF-β and MMP9, did not show significant differences compared to the fibrosis group at day 32. DNC post-treatment should be administered immediately following isoproterenol use or, in medical terms, immediately after a heart attack to be effective.

(Discussion, Page 17, lines 612-614)

Reviewers’ comment: Was there any evidence ISO triggers pulmonary fibrosis?

Response: Isoproterenol has not been used in any previous study to induce pulmonary fibrosis. As explained in the discussion section, bleomycin is typically used through an invasive method to induce this condition. We used isoproterenol for the first time to investigate the induction of pulmonary fibrosis. Important genes such as COL3, SMA, COL1, TGF-β, and MMP9 showed increased expression compared to the control group. However, unlike cardiac fibrosis where gene expression changes were observed on days 5 and 32, in pulmonary fibrosis, these genes showed a scattered expression pattern across different days.

This isoproterenol-induced pulmonary fibrosis model might be used in studies of cardiopulmonary fibrosis, if more decisive method of pulmonary fibrosis induction is not available.

Reviewers’ comment: The comparison of the RNAseq data should be interpreted with caution as the analyses were performed on completely different samples, human fibroblasts stimulated with TGFb in vitro and mice hearts subject to LAD ligation. It would have been conceivable had the authors have used RNAseq generated from human ischemic hearts (which are widely available) vs mice LAD. Also problematic is the stimulation times, 24hs in human vs days/weeks in mice. In addition, the RNAseq analysis is poorly explained and presented.

Response: In the bioinformatics section, our goal was to identify a gene that plays a role in every cardiac fibrosis induction model, including cell culture conditions. As mentioned, we have various methods for inducing cardiac fibrosis and can use different compounds. For this reason, we included different induction methods in this study. We also separately analyzed both human and mouse samples to find a common gene, so that after confirming its changes in mice, we could use it for investigation in humans as well. In fact, our objective was not to find a gene specific to one particular cardiac fibrosis induction model. Instead, we aimed to identify a gene that is universally involved across all conditions. This approach ensures that researchers can observe changes in this gene's expression regardless of the method used to induce fibrosis.

Given that we utilized samples from various studies that shared only cardiac fibrosis as a common factor, with different induction methods and species under investigation, all samples were not analyzed together or pooled. Initially, we analyzed the samples from each project separately and obtained the differentially expressed genes (DEGs) related to those samples.

However, to achieve our stated goal of finding a gene common to all these methods, we ultimately employed Venn analysis. By creating a Venn diagram, we were able to identify genes shared across all these studies. We visualized these identified common genes through a heat map. Additionally, we reported their molecular, biological, and cellular roles using functional enrichment analysis.

The interpretation of the enrichment results was also added to the manuscript. In addition to these aspects, details about the other tools used, such as STRING and Cytoscape, were added in the Methods and Results sections.

(Materials and methods (Eligibility criteria and dataset selection), Page 7, line 245-250)

(Materials and methods (Finding hub genes), Page 8, lines 275-282)

(Results (Identification of DEGs

---

## [Decision Letter · Decision Letter 1]

4 Sep 2024

PONE-D-24-22014R1Preventive and treatment efficiency of Dendrosomal Nano-Curcumin against ISO-induced cardiac fibrosis in mouse modelPLOS ONE

Dear Dr. Soltani,

Thank you for submitting your manuscript to PLOS ONE. After careful consideration, we feel that it has merit but does not fully meet PLOS ONE’s publication criteria as it currently stands. Therefore, we invite you to submit a revised version of the manuscript that addresses the points raised during the review process.

The provision of full heart sections is appreciated; however, the methodology of fibroblast isolation is problematic. I have personal experience isolating WBCs, endothelial cells, vsmcs and fibroblasts from the heart. The methods provided would not exclude vsmc from the culture. In my experience, these method would yield ~50% vsmc. Alpha SMA is a vsmc marker, and vimentin is also expressed in vsmc. Authors need to verify the absence of MYH11 and smoothelin, vsmc-specific markers, to assess the purity of the cells. While PCR would suffice, flow cytometry would be ideal.

We look forward to receiving your revised manuscript.

Kind regards,

Rami Salim Najjar, Ph.D.

Academic Editor

PLOS ONE

Reviewers' comments:

Reviewer's Responses to Questions

**Comments to the Author**

1. If the authors have adequately addressed your comments raised in a previous round of review and you feel that this manuscript is now acceptable for publication, you may indicate that here to bypass the “Comments to the Author” section, enter your conflict of interest statement in the “Confidential to Editor” section, and submit your "Accept" recommendation.

Reviewer #1: All comments have been addressed

2. Is the manuscript technically sound, and do the data support the conclusions?

Reviewer #1: Partly

3. Has the statistical analysis been performed appropriately and rigorously? 

Reviewer #1: I Don't Know

4. Have the authors made all data underlying the findings in their manuscript fully available?

Reviewer #1: Yes

5. Is the manuscript presented in an intelligible fashion and written in standard English?

Reviewer #1: Yes

6. Review Comments to the Author

Reviewer #1: (No Response)

7. PLOS authors have the option to publish the peer review history of their article (what does this mean?). If published, this will include your full peer review and any attached files.

Reviewer #1: No

---

## [Author Response · Author response to Decision Letter 1]

20 Sep 2024

Response to Reviewers

Submission ID: PONE-D-24-22014

Manuscript Title: Preventive and treatment efficiency of Dendrosomal Nano-Curcumin against ISO-induced cardiac fibrosis in mouse model

Dear Editor;

I want to thank the reviewers for their careful reading of this manuscript and their suggestions and thoughtful comments which helped to improve the quality of this manuscript. 

Our response follows:

Reviewers' comment: Authors need to verify the absence of MYH11 and smoothelin, vsmc-specific markers, to assess the purity of the cells. While PCR would suffice, flow cytometry would be ideal.

Response: Unfortunately due to the financial limitations and sanctions against our country, providing specific antibodies against MYH11, and Smoothelin to do flow cytometry is challenging and too expensive. Due to these limitations, we were only able to acquire primers for MYH11 to conduct PCR testing.

For our cell culture experiments, we used the heart tissue without the aorta, which significantly reduced the presence of VSMCs chance. Consequently, this method helped us to minimize the VSMC population in the initial cell cultures. Additionally, as the cells were passaged, the purity of fibroblasts in the culture increased, since VSMCs tend to be outcompeted in such conditions.

We also conducted an RT-PCR test against MYH11 expression on our cultured cells. As is shown in the attached figure, no MYH11 gene-related band was observed, indicating the absence or non-detectable number of VSMCs in our cultured cell preparation. Noteworthy, we used mouse lung tissue as a positive control of the RT-PCR reaction, where MYH11 is consistently expressed in the smooth muscle cells of the pulmonary vasculature and airways. 

The MYH11 primer sequences have been added to the table of primer sequences for use in RT-PCR testing. The negative result of the MYH11 PCR test has been added to the Materials and Methods section of the manuscript.

---

## [Editor Report · Decision Letter 2]

26 Sep 2024

Preventive and treatment efficiency of Dendrosomal Nano-Curcumin against ISO-induced cardiac fibrosis in mouse model

PONE-D-24-22014R2

Dear Dr. Soltani,

We’re pleased to inform you that your manuscript has been judged scientifically suitable for publication and will be formally accepted for publication once it meets all outstanding technical requirements.

Kind regards,

Rami Salim Najjar, Ph.D.

Academic Editor

PLOS ONE
---

## [Editor Report · Acceptance letter]

2 Oct 2024

PONE-D-24-22014R2 

PLOS ONE

Dear Dr. Soltani, 

I'm pleased to inform you that your manuscript has been deemed suitable for publication in PLOS ONE. Congratulations! Your manuscript is now being handed over to our production team.

Kind regards, 

on behalf of

Dr. Rami Salim Najjar 

Academic Editor

PLOS ONE